

# A progressively elevated temperature (PET) IRSL SAR procedure – first experiments and results

Annette Kadereit[1], Mariana Sontag-González[1,2,3], Sebastian Kreutzer[1], Marco Colombo[1], Christoph Schmidt[4,] Paul R. Hanson[5]

[1]Institute of Geography, Heidelberg University, Im Neuenheimer Feld 348, 69120 Heidelberg, Germany
[2]Luminescence Dating Laboratory, Research Laboratory for Archaeology & the History of Art, School of Archaeology, University of Oxford, OX1 3QY Oxford, UK
[3]Institute of Geography, Justus Liebig University of Giessen, 35390 Giessen, Germany
[4]Institute of Earth Surface Dynamics, University of Lausanne, Bâtiment Géopolis, 1015 Lausanne, Switzerland
[5]Conservation and Survey Division, School of Natural Resources, University of Nebraska-Lincoln, Lincoln, Nebraska, USA

*Correspondence to*: Annette Kadereit (annette.kadereit@uni-heidelberg.de)

**Abstract.** Infrared stimulated luminescence (IRSL) dating is a common technique for dating feldspar-bearing sediment deposits. The technique is preferentially applied as a single aliquot regenerative (SAR) protocol derivative: Post-infrared infrared ($pIR_{1st}IR_{2nd}$) and multiple elevated temperature (MET) SAR protocols. Both of these techniques aim to reduce problems with unstable luminescence traps common to feldspars, known as anomalous fading. Elimination of the unstable IRSL signal component, which may lead to age underestimation, is achieved by sequential sampling of the IRSL signal at either two ($pIR_{1st}IR_{2nd}$) or a few (MET) discrete temperatures, usually in the range 50 °C to 290 °C. We propose a modified approach, the progressively elevated temperature (PET) IRSL SAR, which continuously records the luminescence signal while progressively elevating the sample temperature. The benefits of this approach include the generation of quasi-continuous data chains, beginning with the recorded PET-IRSL signal curves, over De-value curves relevant for palaeodose assessment, a-value curves used for dose-rate refinement, and g-value curves serving to assess the signal loss resulting from fading. In addition to anomalous fading problems, feldspars do not bleach as quickly and thoroughly as quartz, potentially resulting in age overestimation. The illustrative PET data curves may be useful in illustrating a sample´s bleaching and fading history. Thus, they may allow users to both reduce problems with anomalous fading and identify aliquots that are least impacted by incomplete bleaching.



## 1 Introduction

Feldspar is a ubiquitous mineral widely used as a dosimeter for age determination of sediment deposits (e.g., Preusser et al., 2008; Tsukamoto, 2025) with luminescence dating techniques (Aitken, 1985b, 1998). Low-level environmental radioactivity ionizes electrons in the crystal, some of which are trapped at lattice defects (electron traps) over time as a function of the local environmental dose rate. As the luminescence traps are emptied by light and the sediment shields the grains from energy input (light, heat), the storage time ideally corresponds to the last cycle of sediment reworking and deposition. For palaeodose determination, the samples are stimulated in the luminescence reader to release the trapped electrons, some of which will recombine with radiative hole-centers emitting luminescence, a proxy of the unknown palaeodose, determined as equivalent dose (De). Feldspar luminescence may be stimulated by infrared (IR) (infrared stimulated luminescence, IRSL, Hütt et al., 1988), and the resultant ages can be determined using a single-aliquot regeneration (SAR) protocol (Murray and Wintle, 2000).

Like quartz, feldspar comes with advantages and disadvantages for luminescence dating. The luminescence traps in feldspars may be unstable and produce age underestimates due to so-called "anomalous fading" (Wintle, 1973). Further, feldspars bleach relatively slowly compared to quartz (Godfrey-Smith et al., 1988), potentially leading to age overestimation. Although these limitations are problematic, dating feldspars with luminescence does have several key advantages. The upper limit for storing a (palaeo)dose is much higher for feldspar than for quartz (e.g., Tsukamoto, 2025), allowing older samples to be dated. The accuracy and precision of feldspar ages are less affected by the external environmental dose rate and changing sediment-water contents due to their relatively high internal dose rate related to their $^{40}K$ and $^{87}Rb$ contents.

Not least due to the fading phenomenon, feldspar has been investigated thoroughly to elucidate the physical properties that result in age underestimation and to mitigate those issues. Depending on the temperature, electrons in the IRSL traps may overcome the energy difference to the conduction band (optical activation energy ca 2.5 eV; Hütt et al., 1988). In addition to this thermally assisted „normal" electron-escape, „anomalous" fading of the luminescence signal may be explained by several mechanisms, such as quantum-mechanical tunnelling (Spooner, 1994; Visocekas, 1985, 1993; Visocekas et al., 1994, 1998), especially at low temperatures and localized transitions (Templer, 1986; Tyler and McKeever, 1988), especially above room temperature. Therefore, next to a temperature-independent fading, there exists a temperature-dependent signal decrease related to the luminescence trap depth (Molodkov et al., 2007; Wintle, 1977). Temperature-dependent fading was explained by electrons „hopping" through band-tail states (Guérin and Visocekas, 2015). Graphical presentations of the complex mechanisms are given by, e.g., Riedesel et al. (2019) and Ankjærgaard and Jain (2010).

To reduce the impact of anomalous fading, the post-infrared infrared-stimulated (pIRIR) protocol was introduced. With the pIRIR-technique, firstly, near-neighbor electron transitions from trap to hole are stimulated by low-temperature IRSL



readout (pIR$_{1st}$), which are interpreted as the remains of a potentially unstable, fading component. Afterwards, the more stable population of trapped electrons is sampled by high-temperature IRSL readout (pIR$_{2nd}$) (Thomsen et al., 2008). While the initial

low-temperature readout usually occurs at 50 °C or 60 °C, the successive high-temperature readout may be performed, e.g., at 280 °C or 225 °C (pIR$_{50}$IR$_{280}$; pIR$_{60}$IR$_{225}$) with associated preheat temperatures (PHT) of, e.g., 320 °C or 280 °C, respectively (Buylaert et al., 2009; Lamothe et al., 2020; Thiel et al., 2011; Thomsen et al., 2011, 2008). The higher the stimulation temperature, the more stable the resulting IRSL signal, but at the cost of signal bleachability (Buylaert et al., 2012). Lower temperatures are usually considered more appropriate for young samples, e.g., of Holocene age (e.g., Reimann et al., 2011;

Reimann and Tsukamoto, 2012). Alternative to two-step IRSL-readouts, multiple-elevated temperature (MET) IRSL readouts are applied, e.g., from 50–250 °C, in five steps of 50 K (e.g., Li and Li, 2011; Schwahn et al., 2023). For younger, Holocene samples, a MET protocol was introduced with modest PHT of 200 °C and successive IR-readout from 50–170 °C in five steps of 30 K, which under favorable conditions may be reduced to a three-step MET (Fu and Li, 2013; cf. also **Supplement 6, Fig. S6.1**). Modest preheat and MET-IRSL readouts may also work for sediments up to ca 500 ka (Gegg et al., 2024). A

considerable advantage of MET protocols is enabling us to check whether De values rise with temperature, which may indicate partial bleaching, or whether they plateau from a certain temperature onwards, which can also be interpreted as signal stability (no fading). For instance, Bateman et al. (2025) focused on aliquots exhibiting De plateaus for age calculation and removed aliquots affected by fading and/or partial bleaching as indicated by the absence of a De plateau.

A logical next step is to develop the two- and multi-step techniques (pIRIR, MET) further to continuously monitor IRSL with rising temperature up to the highest readout temperature to derive a quasi-continuous set of De values. A continuous record might show more clearly **(1)** if a plateau exists (in the higher temperature range); **(2)** whether samples are adequately bleached prior to deposition; **(3)** temperatures best suited for De determination reaching the optimal compromise between low fading and sufficient signal resetting.


Therefore, here we introduce a progressively elevated temperature (PET) IRSL dating procedure, PET-IRSL SAR that uses the well-established SAR protocol (Murray and Wintle, 2000). Similar to the pIRIR and MET procedures, our approach initially eliminates a potentially unstable component of the IR-signal by low-temperature readout before recording a more stable IR-signal during increasingly higher temperatures. The PET procedure shares similarities with the thermally

modulated (TM) procedure developed for quartz OSL dating (Chruścińska et al., 2020; Chruścińska and Szramowski, 2018; Palczewski et al., 2025). However, the TM procedure was developed to gradually increase the optical cross-section of the quartz OSL trap when stimulated by red light. In contrast, the PET-IRSL approach aims primarily at lifting the electrons from the IRSL trap(s) successively to increasingly higher band-tail states to induce electron recombination **(1)** with local/near-neighbor electron-holes via lower band tails in the early phase of the PET-IRSL readout, and **(2)** with increasingly further

distant centers via higher band tails and/or the conduction band in a later phase, as suggested by the feldspar luminescence model (Riedesel et al., 2019).



## 2 Material, instrumentation and software for data analysis

### 2.1 Sample material

100        We use potassium feldspar from three sediment samples of different geographical regions, depositional environments and ages. In addition, we support our observations using three commercially available feldspar specimens with known potassium, sodium and calcium contents.

       **(1)** Sample HDS-1827 represents a fluvial palaeochannel deposit of the former Bergstraßenneckar (BSN) in
southwestern Germany. The sample was taken at the Schäffertwiesen site near Heidelberg from a gravelly-sandy layer (drill core SW04b, 3.15–3.30 m b.g.l.) of likely Late Glacial Age (Engel et al., 2022).

       **(2)** HDS-1776 was collected from the loess-palaeosol section (LPS) Baix in southern France. The sample has been investigated with several OSL techniques, including feldspar coarse-grain screening (Pfaffner et al., 2024) and $pIR_{60}IR_{225}$
dating of feldspar coarse-grains (Pfaffner et al., 2025, in press). The sample originates from the remains of a buried Calcic Cambisol ($B_w$ horizon; IUSS Working Group WRB, 2022), ca 8 m b.g.l. The available publications suggest a palaeodose of ca 220 Gy for the feldspar coarse-grains and a depositional age of ca 55 ka for the sediment in which the palaeosol developed, possibly until ca 40 ka ago.

115        **(3)** Sample HDS-1849 is an aeolian sand sample collected from a sediment exposure ca 2 m b.g.l. in the Nebraska Sand Hills, USA. While the primary dune forms most likely date to the late Pleistocene (Mason et al., 2011), the dunes were reactivated multiple times in the Holocene (Miao et al., 2007). The sample was taken ca 30–40 cm beneath a late Holocene soil. Quartz coarse-grain OSL dating on sample HDS-1849 suggest a deposition age of ca 1 ka (Kreutzer, unpublished data).

120        **(4)** In addition, we applied PET-IRSL SAR measurements to three commercially available feldspar samples that have well-defined elemental concentrations. We analyzed 150–200 μm grains from samples of Norfloat-Potash-Feldspar, G-40-Feldspar and F-20-Feldspar with potassium contents ($KO_2$) of 12.0 wt.%, 10.4 wt.% and 4.1 wt.%, respectively (cf. Kadereit et al., 2020, GChron discussion, Final response).

125        Additional information on the sediment samples and feldspar specimens is given in the **Supplement 1–6**: (1) HDS-1776; (2) HDS-1827; (3) HDS-1849; (4) feldspar specimens; (5) PET-IRSL a-values and g-values of individual aliquots; (6) $pIR_{1st}IR_{2nd}$ and MET SAR measurements, PET-IRSL measurements on empty sample carriers, dose-rate data, and IRSL spectra.



**2.2 Instrumentation**

Luminescence measurements were performed at the Heidelberg Luminescence Laboratory (heiLUM) and at the Giessen Luminescence Laboratory (GLL; spectrometer measurements) using Freiberg Instruments (FI) readers (model *lexsygresearch*; Richter et al., 2013; Freiberg Instruments, 2025) and a Risø reader (model *TL/OSL DA20*; Lapp et al., 2012, 2015) (for details see **Table 1**).


At the time of the PET-IRSL measurements the two FI readers at heiLUM (nicknamed LR03, "Pulse" and LR04, "Colour") had dose rates of ca 7.2 Gy min$^{-1}$ (LR03, "Pulse") and ca 6.4 Gy min$^{-1}$ (LR04, "Colour"). IR and optical stimulation and detection were performed in continuous-wave (CW) mode. IR-stimulation for PET-IRSL occurred at 60 mW cm$^{-2}$ (except few tests at 270 mW cm$^{-2}$); green and blue light stimulated luminescence (GRSL on LR03, "Pulse"; BLSL on LR04, "Colour")

for hotbleaches at 90 mW cm$^{-2}$. Detection of the thermally stimulated luminescence (TL) signal (emitted during sample preheating) and the PET-IRSL signal occurred for the violet-blue feldspar emission (ca 410 nm; henceforth "blue" emission) through a filter package of BG39 (Schott, 1 mm) and AHF-410/40-ET-Bandpass (Chroma, 3.5 mm) for PET-IRSL, supplemented for TL on LR04 ("Colour") by KG3 (Schott, 3 mm). GRSL and BLSL hotbleach signals were detected around 365 nm through a filter package of U340 (Hoya, 2.5 mm) and BP-362/25 (Delta, 2 mm). For sample bleaching, the LED-based

bleaching facility (solar simulator) of the lexsyg readers was used. Luminescence spectra were measured on an FI reader at GLL (nicknamed "Gauss").

Additional measurements with an IR-stimulated MET protocol (Fu and Li, 2013) on sample HDS-1827 and a pIR$_{60}$IR$_{280}$ protocol on HDS-1776 were performed on the Risø luminescence reader (nicknamed "Athenaeum"). IR stimulation

was performed at 90 % power (ca 36 mW cm$^{-2}$) after 10 s warm-up. The blue feldspar emission around 410 nm was detected through an interference filter CH-30D410-44.3 (Chroma) (cf. **Supplement 6, Fig. S6.1, Fig. S6.2.1–S6.2.3**).

For alpha-irradiation, an ELSEC Alpha-Irradiation Unit Type 721 S/N6 with six calibrated $^{241}$Am sources (ca 4.06 Gy min$^{-1}$; Singhvi and Aitken, 1978) was used.


Radionuclide contents of samples HDS-1827 and HDS-1849 needed for environmental dose-rate assessment were determined at heiLUM on 3 g of finely ground sediment with the µDose system (serial no. 26, nicknamed "Trick") (for details see Kolb et al., 2022; Tudyka et al., 2018, 2020). Sediment grinding was performed in a ball mill (Retsch MM 400; frequency of 29.5 Hz for 45 min).




**Table 1.** Reader settings and components.

| Parameter | Settings | | | |
|---|---|---|---|---|
| *Device* | | | | |
| *Name* | LR03 ("pulse") Heidelberg | LR04 ("colour") Heidelberg | Athenaeum Heidelberg | Gauss Gießen |
| *Model* | lexsygresearch | lexsygresearch | OSL/TL DA-20 | lexsygresearch |
| *Manufacturer* | Freiberg Instruments | Freiberg Instruments | Risø DTU | Freiberg Instruments |
| **Irradiation** | | | | |
| *Name* | $^{90}Sr/^{90}Y$ | $^{90}Sr/^{90}Y$ | $^{90}Sr/^{90}Y$ | $^{90}Sr/^{90}Y$ |
| *Init. Activity* | 1.85 GBq | 1.85 GBq | 1.48 GBq | 1.6 GBq |
| *Dose rate* | $0.121 \pm SE$ (0.003) Gy s$^{-1}$ ($\sim$ 7.3 Gy min$^{-1}$) | $0.109 \pm SE$ (0.004) Gy s$^{-1}$ ($\sim$ 6.5 Gy min$^{-1}$) | $0.081 \pm SE$ (0.004) Gy s$^{-1}$ ($\sim$ 4.8 Gy min$^{-1}$) | $0.059 \pm SE$ (0.003) Gy s$^{-1}$ ($\sim$ 3.5 Gy min$^{-1}$) |
| *Cal. material* | LexCal #3Gy_0050 (90-150 µm) | LexCal #3Gy_0050 (90-150 µm) | QQ5 quartz (90-150 µm) | Risø calibration quartz batch #200 6.0 Gy (180−250 µm) |
| *Cal. date* | 2024-03-11 | 2024-05-14 | 2021-03-08 | 2022-10-15 |
| **Detection** | | | | |
| *Detector* | ET Enterprise 9235QB04 | Hamamatsu H7360-02 | EMI 9235QB15 | Andor Shamrock 163 + Andor Newton DU920P + 2.4 mm quartz fibre 300 l/mm Blaze 500 nm |
| *Filter blue/violet* | Schott BG39 (1 mm) Chroma 410/40 ET (3.5 mm) | Schott BG39 (1 mm) Chroma 410/40 ET (3.5 mm) | Chroma CD-30D410-44.3 | - |
| *Filter UV* | Hoya U340 (2.5 mm) Delta BT-362/25 (2 mm) | Hoya U340 (2.5 mm) Delta BT-362/25 (2 mm) | - | - |
| *Filter heat protection* | - | Schott KG3 (3 mm) | - | - |
| *Filter IRSL spectra* | - | - | - | Schott BG39 (3 mm) |
| **Stimulation** | | | | |
| *Blue* | - | LEDs 458 $\Delta$ 3 nm (90 mW cm$^{-2}$) | - | - |
| *Green* | LDs 523 nm (90 mW cm$^{-2}$) | - | - | - |
| *IRSL* | LDs 850 nm (60 mW cm$^{-2}$) | LEDs 850 nm (60 mW cm$^{-2}$) | LEDs 870 $\Delta$ 40 nm (40 mW cm$^{-2}$) | LDs 850 nm (250 mW cm$^{-2}$) |
| **Bleaching** | In-built solar simulator | In-built solar simulator | | |
| **Software** | LexStudio2 v2.28.6 | LexStudio2 v2.28.6 | Sequence Editor v4.74 | LexStudio2 v1.7.4 |

## 2.3 Sample preparation

For HDS-1776 and HDS-1849, sample preparation followed standard procedures for extracting 125–212 µm
potassium feldspar (cf. Pfaffner et al. 2025, in press, Fig. S2.2.1). HDS-1827 underwent minimal sample preparation,
delivering polymineral coarse-grains for profile screening (Pfaffner et al., 2024).



Mineral separates were settled on stainless steel cups (ca 10 mm in diameter, ca 1 mm thick) with silicon oil. Two types of aliquots were prepared: **(1)** "small aliquots" (a few hundred grains each) with a 4-mm mask for dim samples (HDS-1827, HDS-1849) and **(2)** "tiny aliquots" (a few tens of grains) with a 2-mm mask for bright samples (HDS-1776; feldspar specimens). Additionally, **(3)** few single-grain (SG; Galbraith et al., 1999) aliquots of HDS-1776 were prepared on sample carriers designed for multiple-grain analysis.

## 2.4 Software for data analysis

For luminescence data analysis, we applied the R (R Core Team, 2025) package "Luminescence" (Kreutzer et al., 2025a). De-value assessment was done with the function "analyse_SAR.CWOSL()" by applying a single-exponential fit and ignoring any rejection criteria. The g-value determination followed Huntley and Lamothe (2001) and was performed with the function "analyse_FadingMeasurement()". Age corrections were calculated with the function "calc_FadingCorr()". For additional analyses, we also used the function analyse_pIRIRSequence() (cf. **Supplement 6, Fig. S6.3**) and the software "Luminescence Analyst" (Duller 2015) (cf. **Supplement 6, Fig. S.6.1, Fig. S6.2.1–6.2.3, Fig. S6.4.1–6.4.2**).

The spectral measurements were smoothed to remove abrupt signal spikes by applying a median of length 3 along the temperature axis and then along the wavelength axis iteratively 6 times using the function "apply_CosmicRayRemoval()".

## 3 The progressively elevated temperature (PET) IRSL approach

Loess sample HDS-1776 served to develop the PET-IRSL technique. Its wider applicability was tested on the fluvial sediment sample HDS-1827 and the dune sample HDS-1849, and to a lesser extent on the three feldspar specimens. Additionally, we investigated whether plausible De values may be gained with the new approach for the three sediment samples. However, the aim of the present study is not the dating of the samples but to test the PET-IRSL approach.

## 3.1 The PET-IRSL SAR protocol

The PET-IRSL SAR approach intends to provide a further development of the commonly used IRSL SAR, $pIR_{1st}IR_{2nd}$ SAR and MET SAR dating techniques (e.g., Juschus et al., 2007; Thomsen et al., 2008; Li and Li, 2011). For a thermally assisted IR-readout from 60 °C to 280 °C, several parameters may be adapted, such as, the temperature ramp and the IR stimulation power at the sample position. After several test measurements (not shown here) we opted for 60 mW cm$^{-2}$ stimulation power at the sample position (see upper level of the green-shaded area with respect to right y-axis in **Fig. 1**) and a heating rate of 2 K/s for the ramp from 60 °C to 280 °C (see red solid line and 180–290 s x-axis range in **Fig. 1**). To eliminate the unstable fading component, the initial IR-readout at 60 °C was administered for 120 s (low-temperature PET-IRSL range),



before elevating the temperature by 2 K/s for the following high-temperature PET-IRSL readout. The high-temperature PET-IRSL readout follows immediately after the 120 s 60 °C low-temperature PET-IRSL readout, which contrasts with $IR_{1st}IR_{2nd}$ and MET techniques. The preheat always lasted 60 s, whereas the preheat temperature varied. Hereinafter, a specific preheat procedure is abridged as, e.g., PHT 320 °C. The detection time (0–335 s; cf. cross-hatched area in **Fig. 1**) includes an initial warmup and temperature stabilization at 60 °C (red line 0–60 s), the actual PET-IRSL readout (red line and green-shaded area 60–290 s), and a cooldown after PET-IRSL readout and another 25 s at 60 °C elevated temperature (red line 290–335 s).


The preheat and maximum stimulation temperatures ($PET_{max}$) were adjusted according to the expected sample age. For HDS-1776 from the LPS Baix, the oldest sample in the data set, we used PHT 320 °C and $PET_{max}$ 280 °C. For the two younger samples, a lower preheat temperature and a lower $PET_{max}$ were adopted (cf. tests in **sections 4.1–4.3**). The length of the measurement and detection time were the same for all PET-IRSL measurements. We only decreased $PET_{max}$, which in

effect leads to a lower ramp of 1 K/s for $PET_{max}$ 170 °C as applied, e.g., for De measurements on samples HDS-1827 (BSN) and HDS-1849 (Nebraska) after PHT 200 °C.

The PET-IRSL step is embedded in the SAR protocol between a preceding PHT step and a subsequent hotbleach (HBL) step. The hotbleaches were performed for 250 s using GRSL at 90 mW cm$^{-2}$ on LR03 ("Pulse") or BLSL at

90 mW cm$^{-2}$ on LR04 ("Colour"). PHT was performed for 60 s as a TL step.

Dose-response curve construction (**Fig. 2**) follows a scheme as often applied at heiLUM: Six regeneration-dose points (REGs) build up the dose-response curve around a 100 % regeneration-dose point (REG 100 %), i.e., an expected dose (natural, NL) or administered laboratory dose (LAB). The dose-response curve is constructed by REGs at 40 %, 70 %, 100 %, 130 %,

160 %, 0 % (test of signal recuperation), 100 % (testing the recycling ratio) (step 8 in **Fig. 2**). For the dose-recovery tests and De-value determination REG 100 % was set to 2500 s beta-irradiation time (ca 300 Gy) for HDS-1776, 1000 s (ca 120 Gy) for HDS-1827 and 100 s (ca 10.6 Gy) for HDS-1849. The different dose-response curves are denoted as "2500 s/300 Gy dose-response curve", "1000 s/120 Gy dose-response curve", or "100 s/10.6 Gy dose-response curve", respectively. Expected doses or REG 100 %, respectively, were derived for HDS-1776 from Pfaffner et al. (2024) and Pfaffner et al. (2025, in press), for

HDS-1827 from a MET test measurement (cf. **Supplement 6, Fig. S6.1**) and for HDS-1849 from the unpublished OSL data by Kreutzer, but were in each case broadly enlarged to gain luminescence counts adequate for data analysis.

The structure of the PET-IRSL SAR protocol is depicted in **Fig. 2**. The stated parameter values are those used after the testing phase for De-value measurements of the three sediment samples (cf. **section 7.2; Supplements 1–3, Fig. S1.5.1,**

**Fig. S2.14, Fig. S3.20**), while other parameter values were checked in pretests on the sediment samples and applied for the three commercial feldspar specimens (cf. **section 4; Supplement 4, Fig. S4.1.1–S4.3.2**).



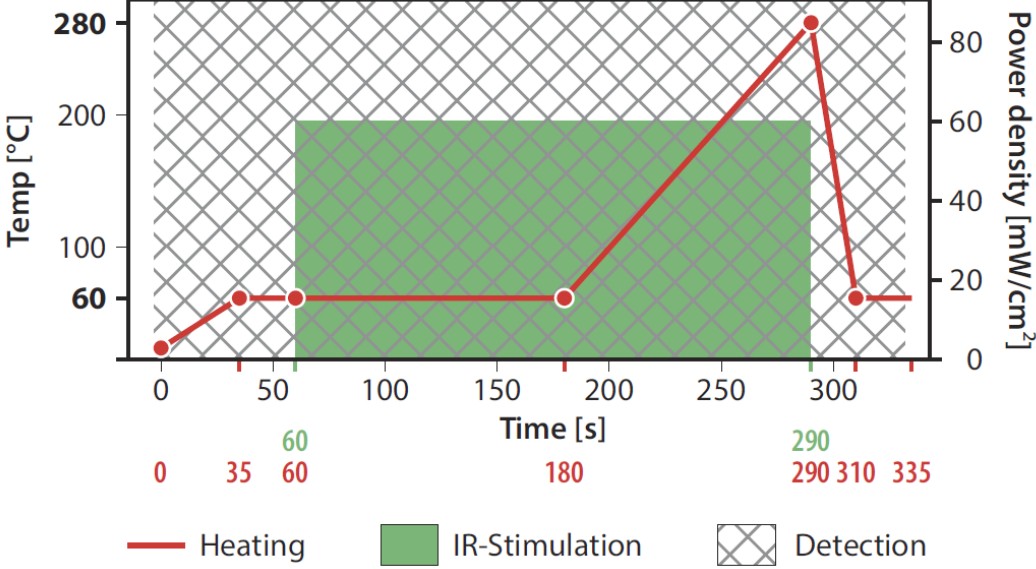

**Figure 1.** The PET-IRSL measurement step included in the PET-IRSL SAR protocol (cf. **Fig. 2**) in the present study on the lexsyg readers. Customized thermal assistance (red line) up to PET$_{max}$ 280 °C (left y-axis) with superimposed IR-stimulation (green box) at 60 mW cm$^{-2}$ LD power density at the sample position (right y-axis) at 60–290 s measurement time (x-axis). Graphical representation following that of the LexStudio2 Sequence Editor (Richter and Kumar, 2024). For further details, see the main text.

Each SAR cycle consists of two parts, the natural or regenerative dose half-cycle and the normalization dose half-cycle. The PET-IRSL SAR protocol differs from the classic SAR protocol (Murray & Wintle 2000) in two aspects: **(1)** OSL is replaced by the PET-IRSL step; **(2)** both the first and the second SAR half-cycle possess a hotbleach step. Thus, each half-cycle consists of four steps (**Fig. 2**): **(1)** beta irradiation (zero for the natural); **(2)** preheat for 60 s at 320 °C or 200 °C; **(3)** customized PET-IRSL 60–280 °C or 60–170 °C; and **(4)** GRSL (LR03, "Pulse") or BLSL (LR04, "Colour") hotbleach for

250 s at 280 °C or 170 °C. Each natural or regeneration half-cycle is followed by a similarly structured normalization-dose half-cycle, in which a constant normalization dose (NRM; also called test dose) is applied to monitor and correct for a sample's sensitivity changes during a complete SAR measurement (normalization procedure after Murray and Wintle, 2000). NRM was 20 % (500 s; ca 60 Gy) for HDS-1776 as transferred from Pfaffner et al. (2025, in press) and verified by an NRM test applying the PET-IRSL SAR approach (cf. **section 4.1**); and 10 % for both HDS-1827 (100 s; ca 12 Gy) and HDS-1849 (10 s; ca 1 Gy)

as determined by respective NRM tests (cf. **section 4.1**). **Fig. 2** and other figures in the main text contain both the beta-irradiation time (s) and the corresponding deposited dose (Gy), while the supplementary only gives beta-irradiation times.



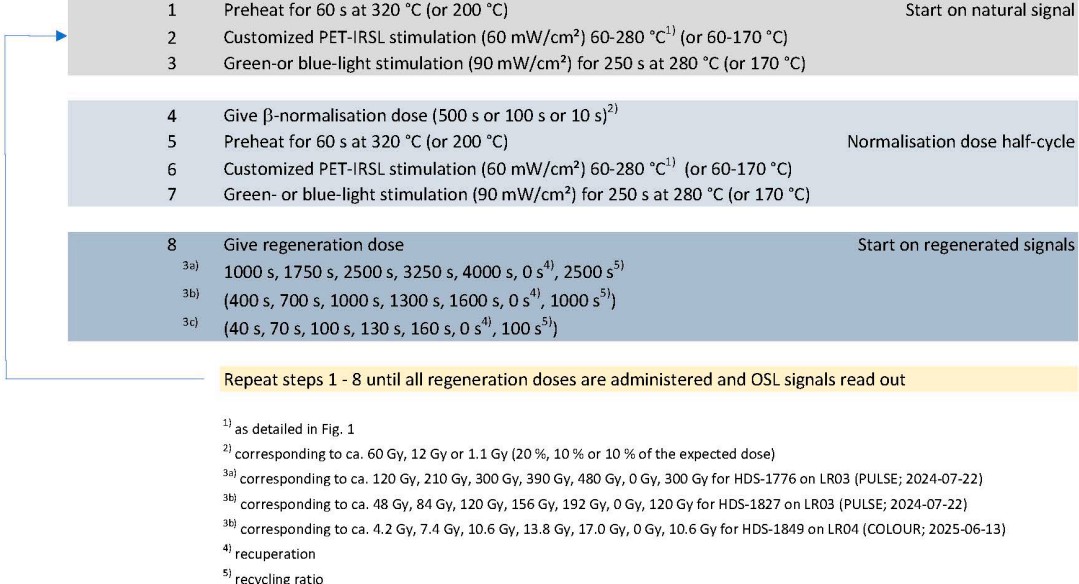

**Figure 2.** Flow chart of the PET-IRSL SAR protocol used on the loess sample HDS-1772, and values in brackets were used for the fluvial sample HDS-1827 and the dune sample HDS-1849. GRSL hotbleaches only on reader LR03 ("Pulse"; samples HDS-1776 and HDS-1827). BLSL hotbleaches only on reader LR04 ("Colour"; sample HDS-1849). Step 8 shows the three different dose-response curves used for the three sediment samples. Regeneration-dose points (REGs) at 40 %, 70 %, 100 %, 130 %, 160 % and 0 % of an expected dose corresponding to REG 100 % and eponymous for a specific dose-response curve: Expected dose 2500 s/300 Gy, "2500 s/300 Gy dose-response curve" applied to HDS-1776. Expected dose 1000 s/120 Gy, "1000 s/120 Gy dose-response curve" used for HDS-1776. Expected dose 100 s/10.6 Gy, "100 s/10.6 Gy dose-response curve" for HDS-1849.

## 3.2 The PET-IRSL signal curve

A typical PET-IRSL signal curve for the loess sample HDS-1776 up to $PET_{max}$ 280 °C after PHT 320 °C is given in **Fig. 3**. Before and after IR is switched on (0–60 s and 290–335 s measurement time), a very low background signal is recorded at 60 °C. When IR-stimulation is switched on (at 60 s), an IRSL shine-down is observed, which after 120 s (180 s total measurement time) reaches the lowest level of luminescence counts (end of low-temperature PET-IRSL range; cf. horizontal yellow bar below top axis). When temperature-ramping 60–280 °C begins (start of high-temperature PET-IRSL range; cf. horizontal orange bar), the signal increases to eventually form a downward open parabola-like curve, the peak of which is positioned at ca 260 s total measurement time or 200 s PET-IRSL readout, respectively, at ca 220 °C. Henceforth, we call the peak of signal counts in the high-temperature range: "PET-peak". When IR-stimulation is switched off and the assisting temperature drops to 60 °C after 290 s total measurement time or 230 s PET-IRSL readout, the signal quickly drops to the background level.





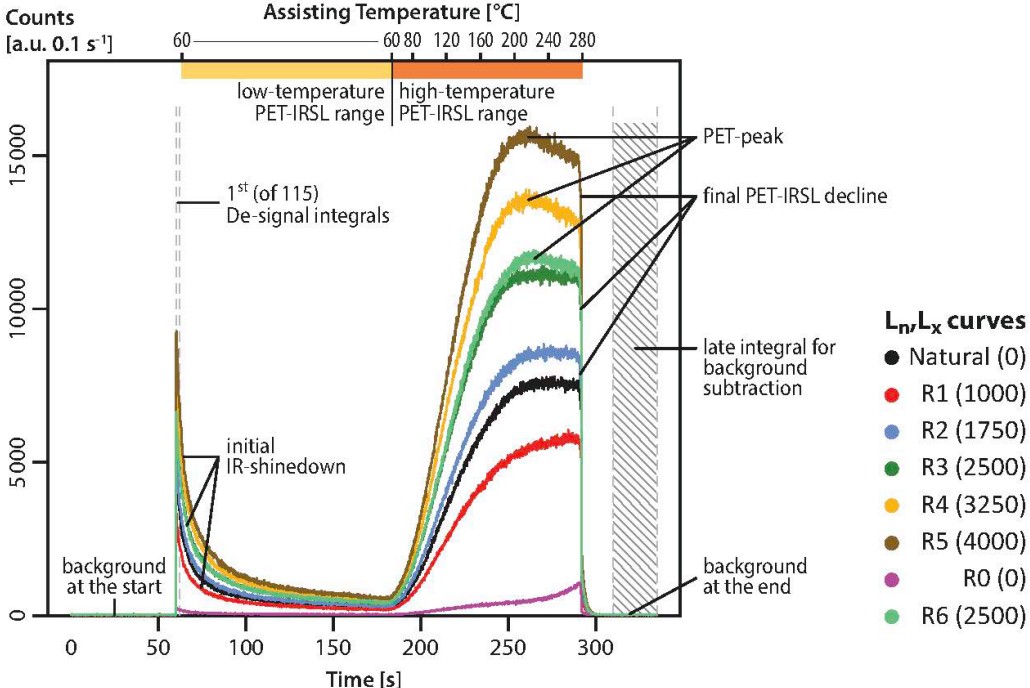

**Figure 3.** PET-IRSL signal curves. PET-IRSL signal curves from eight SAR cycles with natural signal (Ln; black line; third from below) plus seven regenerated signals (Lx; other colors), which include a repeatedly measured regeneration-dose point (dark green and light green) and a zero-dose signal (dark pink). 335 s total measurement time (cf. x-axis). 0–35 s measurement time: temperature increase 25–60 °C. 35–60 s measurement time: temperature kept at 60 °C. 60–180 s measurement time: $IR_{60}$. 180–290 s measurement time: $IR_{60–280}$. 290 s: IR off. 290–315 s: temperature decrease 280–60 °C. 315–335 s: temperature kept at 60 °C. 0.1 s per data channel. 20 data channels per De integral. Dashed grey vertical lines denote the De integral and the background integral (with shaded area in between the lower and upper boundary) used for the De analysis. Graph as exported by the R package "Luminescence" function "analyse_SAR.CWOSL" (slightly modified).

### 3.3 The PET-IRSL SAR De-value curve

For De-value determination, the usual SAR data analysis was applied. The count data of the natural luminescence

signal (Ln) and the luminescence signals (Lx) generated by the regeneration doses were normalized (Ln/Tn, Lx/Tx) according to a sample's sensitivity changes as monitored by the luminescence signal (Tn, Tx) originating from the normalization dose (Murray and Wintle, 2000).

De determination was performed continuously during IR stimulation (60–290 s in **Figs. 1 and 3**) in integrals of 2 s

(20 data channels each; 115 De-signal integrals altogether). The late integral 310–335 s was used for background subtraction. We also applied the PET-IRSL SAR protocol with $PET_{max}$ 170 °C and 280 °C to empty sample carriers to investigate whether



these show an increased background for 60–290 s measurement time, i.e., with elevated temperature and IR switched-on, but could not observe any increase compared to the background 310–335 s after cooldown and IR-switch-off (cf. **Supplement 6, Fig. S6.5a–S6.5b**).


The De values (s irradiation time/Gy; y-axis) were plotted against the De-signal integrals (bottom x-axis) and the corresponding measurement temperature (top x-axis), marking the 60 °C low-temperature PET-IRSL range by a yellow horizontal bar below the top x-axis and the high-temperature PET-IRSL range (ramp 60–280 °C) in orange (**Fig. 4**). All 115 De values form a PET-IRSL De-value curve. Due to the decreasing signal of the initial IR-shine-down (cf. **Fig. 3**), De values

of the late low-temperature PET-IRSL range (De-signal integrals ca 20–60) show relatively high data errors and scatter (cf. also a-value and g-value curves in **sections 5.1–5.2**). The approximate position of the PET-peak (here De-signal integral ca 100 at ca 220 °C), as present for samples measured with $PET_{max} \geq$ ca 240 °C, is marked by a star (**Fig. 4**).

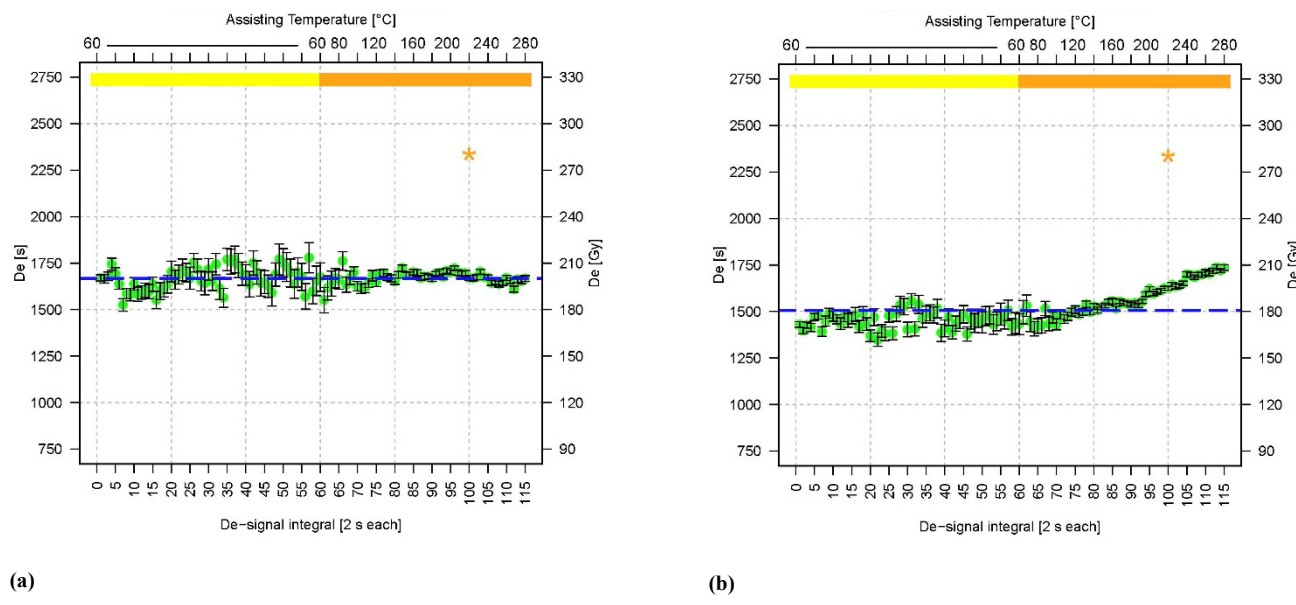

(a)                                                                                     (b)

**Figure 4.** PET-IRSL De-value curves from two aliquots of sample HDS-1776. The PET-IRSL Des were gained with a 2500 s/300 Gy dose-response curve, NRM 20 %, PHT 320 °C, $PET_{max}$ and HBL 280 °C. The dashed blue line represents the mean of the entire dataset. Position of the PET-peak marked by the star symbol.

The PET-IRSL SAR De-value curves show different shapes (**Fig. 4**). In **Fig. 4a,** all De values scatter statistically around a mean value of ca 1670 s (200 Gy). In **Fig. 4b,** De values in the early part of the low-temperature PET-IRSL start around ca 1445 s (173 Gy; first 20 De integrals), exhibiting perhaps a slight initial rise from ca 1417 s (170 Gy; first four De-signal integrals) to ca 1450 s (174 Gy; next 15 De integrals). Overall, the De values of the low-temperature PET-IRSL plateau below the mean, while De values of the high-temperature PET-IRSL rise above the mean, steadily up to ca 1730 s (208 Gy)

for $PET_{max}$ 280 °C.





## 4   Adapting the PET-IRSL SAR procedure – Tests and results

### 4.1   Testing the dose recovery of an administered laboratory dose

For dose-recovery tests (DRT) aliquots were bleached setting the six LEDs of the solar simulator as follows: ultra-violet (UV, 365 nm, 10 mW cm⁻²), blue (462 nm, 63 mW cm⁻²), green (523 nm, 54 mW cm⁻²), yellow (590 nm, 37 mW cm⁻²),

red (625 nm, 105 mW cm⁻²) and IR (850 nm, 90 mW cm⁻²). The procedure closely follows Frouin et al. (2015) but performs the sample bleaching at ambient temperature. If not otherwise stated (cf. PBL tests with white and monochromatic light in **section 6.2**), these settings were applied for one hour before an aliquot was subjected to a DRT, a zero-dose test or an NRM test. Afterwards, the samples received a laboratory dose (LAB) initially corresponding to REG 100 % of the respective dose-response curve (see **Fig. 2**, step 8). Thus, sample HDS-1776 was dosed with LAB 2500 s (300 Gy), sample HDS-1827 with

LAB 1000 s (120 Gy) and sample HDS-1849 with LAB 100 s (10.6 Gy).

As, however, for the analysis of feldspar with IR-stimulation, the size of NRM has a decisive impact on the shape of the dose-response curve and hence on a correct dose recovery (Colarossi et al., 2018), we performed DRTs with varying NRM (NRM tests). These tests represent combined normalization-dose and dose-recovery tests.


For the oldest sample HDS-1776, measured with the 2500 s/300 Gy dose-response curve (see step 8 in **Fig. 2**), we tested NRM 5 %, 7.5 %, 10 %, 20 %, 30 % and 40 % of 2500 s/300 Gy (cf. **Supplement 1, Fig. S1.1.1–S1.1.2**) with the "hot" SAR parameter values PHT 320 °C and HBL 280 °C. All NRM tests recovered LAB 2500 s/300 Gy within a 10 % error margin of the administered dose, with larger NRMs of 30 % and 40 % showing a tendency of underestimation along the lower

limit of the allowed error margin (cf. **Supplement 1, Fig. S1.1.1–S1.1.2**). The best results were gained for NRM 10 % and 20 % (**Fig. 5**). The mean and the expected value are indistinguishable for NRM 10 % (red line overplotting dashed blue line), while the mean for NRM 20 % (dashed blue line) slightly underestimates the expected value (red line). The slightly lower mean is due to De values of the low-temperature PET-IRSL range starting around ca 2400 s (288 Gy), which is ca 4 % below LAB, whereas the high-temperature De values are in good agreement. We decided on NRM 20 % for further measurements

on HDS-1776, as a higher NRM usually leads to lower De-value errors and data scatter.



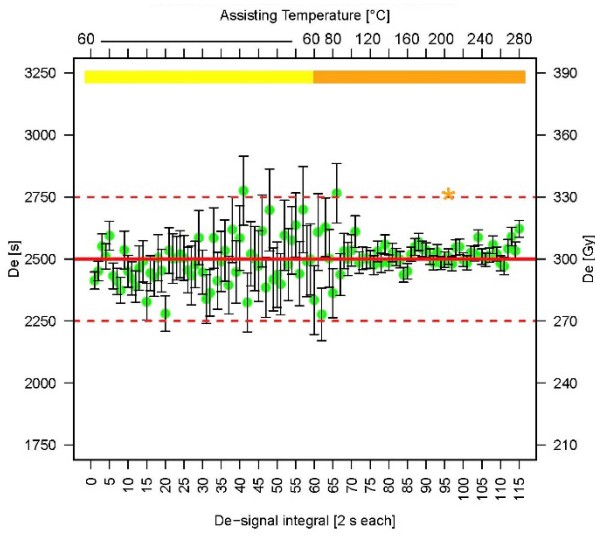

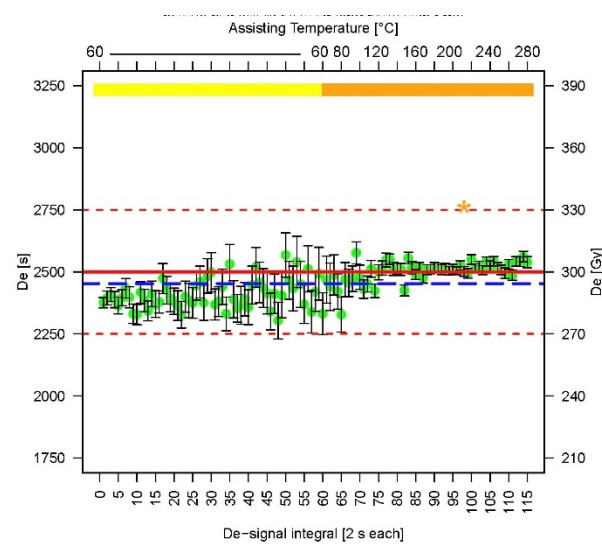

**(a)** NRM 250 s (10 % of 2500 s dose-response curve)     **(b)** NRM 500 s (20 % of 2500 s dose-response curve)

**Figure 5.** NRM test on sample HDS-1776 measured with a 2500 s/300 Gy dose-response curve, LAB 2500 s/300 Gy, PHT 320 °C, $PET_{max}$ and HBL 280 °C, here with **(a)** NRM 10 % and **(b)** 20 % of REG 100 % 2500 s/300 Gy. The expected value is denoted by the solid red line, 10 % error margin by dashed red lines, the mean of all recovered De values by the dashed blue line, position of the PET-peak by the star symbol. For further NRM tests, see **Supplement 1, Fig. S1.1.1**.

A sequence of dose-recovery tests with the "hot" parameter values PHT 320 °C and $PET_{max}$ and HBL 280 °C was also performed on the two younger samples, HDS-1827 and HDS-1849. For these two samples, smaller LAB 1000 s/120 Gy and LAB 100 s/10.6 Gy with a corresponding smaller "1000 s/120 Gy dose-response curve" and "100 s/10.6 Gy dose-response curve" were used (see **Fig. 2**, step 8). Two typical results, one for each sample, are shown in **Fig. 6a** and **Fig. 6b**. Both measurements significantly underestimate the given dose for low-temperature PET-IRSL and meet it at high temperatures around and/or beyond the PET-peak. While HDS-1827 shows a longer plateau of De values (ca 30 De-signal integrals, 160–280 °C), HDS-1849 possesses a modest plateau (< 10 De-signal integrals) around the PET-peak and beyond.

Further measurements on HDS-1827 and HDS-1849 testing different normalization doses with the "hot" SAR parameter values PHT 320 °C and $PET_{max}$ and HBL 280 °C all show similar patterns, i.e., De-value underestimation in the low-temperature range and a high-temperature De-value plateau at a higher level (see **Supplement 2, Fig. S2.1–S2.2, S2.6; Supplement 3, Fig. S3.1–S3.3**).

Dose-recovery using the same 1000 s/120 Gy dose-response curve (HDS-1827) and 100 s/10.6 Gy dose-response curve (HDS-1849) worked best for the two younger samples using PHT 200 °C with $PET_{max}$ and HBL 170 °C. With these





modest thermal parameter values and NRM 10 %, the results fall within an error margin of ca 10 %. The tests were performed

for different LAB (1000 s/120 Gy and 400 s/48 Gy for HDS-1827; 100 s/10.6 Gy, 35 s/3.7 Gy and 15 s/1.6 Gy for HDS-1849;

**Supplement 2, S2.12**; **Supplement 3, Fig. S3.10f, S3.19, S3.22**). One example of each sample, HDS-1827 and HDS-1849, is

illustrated in **Fig. 6c and Fig. 6d.**

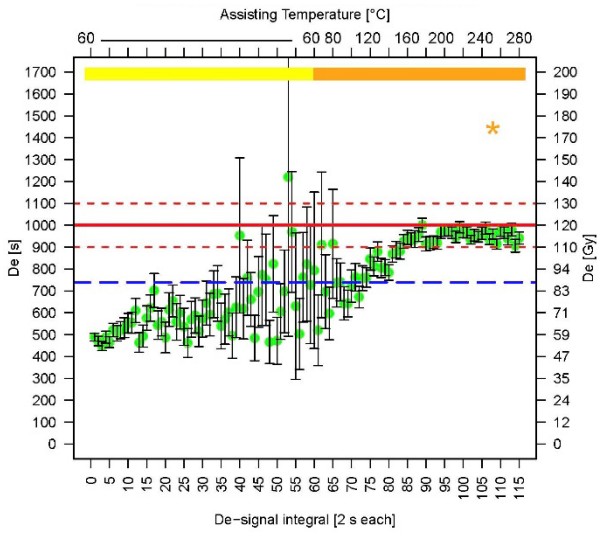

**(a)** HDS-1827 – PHT 320 °C – PET$_{max}$ 280 °C – NRM 30 % of 1000 s dose-response curve, LAB 1000 s

**(b)** HDS-1849 – PHT 320 °C – PET$_{max}$ 280 °C – NRM 40 % of 100 s dose-response curve, LAB 100 s

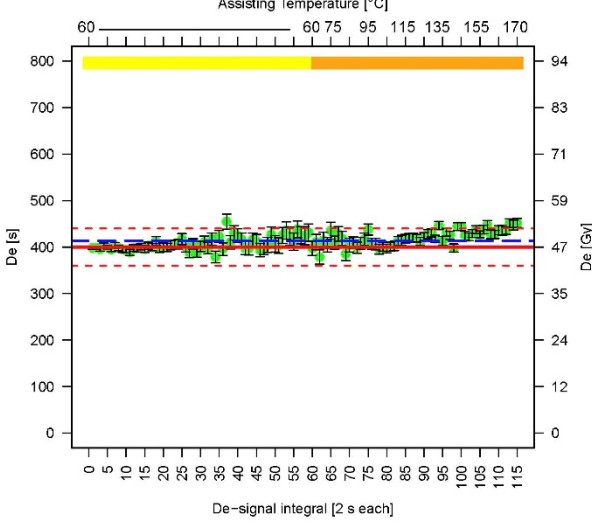

**(c)** HDS-1827 – PHT 200 °C – PET$_{max}$ 170 °C – NRM 10 % of 1000 s dose-response curve, LAB 400 s

**(d)** HDS-1849 – PHT 200 °C – PET$_{max}$ 170 °C – NRM 10 % of 100 s dose-response curve, LAB 100 s





**Figure 6.** NRM tests on samples HDS-1827 (BSN) measured with a 1000 s/120 Gy dose-response curve and HDS-1849 (Nebraska) measured with a 100 s/10.6 Gy dose-response curve (see **Fig. 2**, step 8). **(a, b)** "Hot" SAR parameter values PHT 320 °C, PET$_{max}$ and HBL 280 °C. **(c, d)** "Modest" thermal SAR parameter values PHT 200 °C, PET$_{max}$ and HBL 170 °C. **(a)** HDS-1827 with LAB 1000 s/120 Gy and NRM 30 % (300 s/36 Gy). **(b)** HDS-1849 (Nebraska) with LAB 100 s/10.6 Gy and NRM 40 % (40 s/4.2 Gy). **(c)** HDS-1827 with LAB 400 s/48 Gy and NRM 10 % (100 s/12 Gy). **(d)** HDS-1849 with LAB 100 s/10.6 Gy. The expected value is denoted by the solid red line, 10 % error margin by dashed red lines, the mean of all recovered De values by the dashed blue line, position of the PET-peak by the star symbol (only for "hot" SAR parameter values in **(a)** and **(b)**). Please note that the high-PHT SAR version is not considered the most appropriate one for these samples and was not used for De determination.

## 4.2 Zero-dose tests

We performed PET-IRSL SAR measurements on aliquots, which, like those subjected to DRTs or NRM tests, were bleached in the lexsyg reader for one hour but did not receive a laboratory dose. These zero-dose tests served to determine the residual dose after bleaching. The results for the loess sample HDS-1776 and the fluvial sample HDS-1827 are shown in **Fig. 7**. Sample HDS-1776 (**Fig. 7a**), measured with PHT 320 °C and PET$_{max}$ 280 °C, shows for the first 15 De-signal integrals of the low-temperature PET-IRSL range De values which scatter from close to zero to ca 40 s (5 Gy), while the values in the high-

temperature PET-IRSL approach the PET-peak range around ca 100 s (12 Gy). The recovered De values of HDS-1827 (**Fig. 7b**) range around ca 2–3 s (ca 0.3 Gy), with values scattering above and below, over both the low- and high-temperature PET-IRSL ranges with the latter only going up to PET$_{max}$ 170 °C after PHT 200 °C. The De-signal integrals > 95 give De values with a central value below zero for which the function "analyse_SAR.CWOSL()" returns NA (not available) values. Overall, the residual doses for these samples are small relative to their paleodoses (see **Fig. 16 and 17**), indicating the protocol's

appropriateness for dating.




The results of zero-dose tests on the dune sample HDS-1849 are provided in **Fig. 8**.

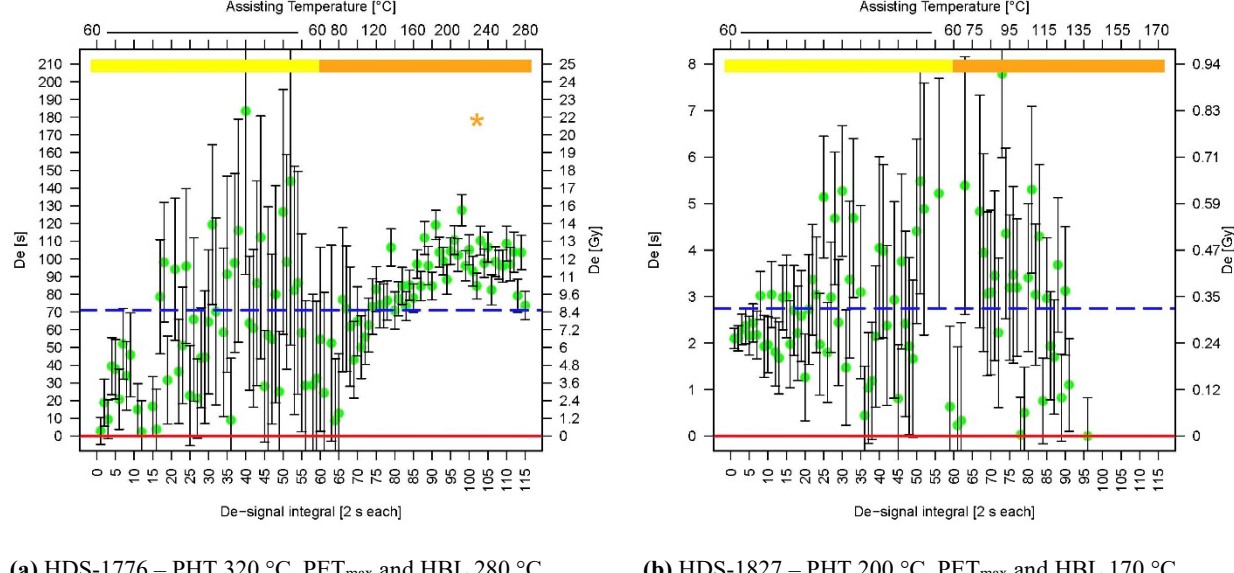

**(a)** HDS-1776 – PHT 320 °C, PET$_{max}$ and HBL 280 °C      **(b)** HDS-1827 – PHT 200 °C, PET$_{max}$ and HBL 170 °C

**Figure 7.** Zero-LAB DRTs on sample **(a)** HDS-1776 from the LPS Baix in southern France with the PET-IRSL SAR protocol with: IR 60 mW cm$^{-2}$, 2500 s dose-response curve, NRM 20 % (500 s), PHT 320 °C, PET$_{max}$ and GRSL hotbleach in each subset of a SAR cycle 280 °C; **(b)** HDS-1827 from fluvial deposits of the Bergstraßenneckar in southwestern Germany, with the PET-IRSL SAR protocol with: IR 60 mW cm$^{-2}$, 1000 s dose-response curve, NRM 10 % (100 s), PET$_{max}$ and GRSL hotbleach in each half-cycle of a SAR cycle 170 °C. Data values of the De intervals > 95 cannot be plotted, as the position of the center value for the late intervals of the high-temperature PET-IRSL range is < 0, for which the function "analyse_SAR.CWOSL()" of the R package "Luminescence" returns NA values.


### 4.3 Optimizing PET$_{max}$ and HBL

We investigated which PET$_{max}$ and HBL may be most appropriate for the aeolian sand sample HDS-1849. We tested 160 °C, 170 °C and 180 °C in a DRT with LAB 35 s/3.7 Gy (cf. **Supplement 3, Fig. S3.19**) and in a zero-dose test (**Fig. 8**). The highly sensitive responding De-value curves suggested PET$_{max}$ and HBL 170 °C as most appropriate. For PET$_{max}$ and

HBL 180 °C, the recovered De values of the high-temperature PET-IRSL increase steadily up to > 3 s (0.3 Gy), which, for the low natural doses of HDS-1849 (see **Fig. 18**), may become relevant. The residual dose increases further with increasing PHT (cf. **Supplement 3, Fig. S3.17**), reaching ca 50 s/ca 5.3 Gy for PHT 320°C, which is a multiple of the low natural doses obtained with the PET-IRSL SAR protocol. We therefore selected PET$_{max}$ and HBL 170 °C with PHT 200 °C for analyzing HDS-1849 and for the fluvial sample HDS-1827.






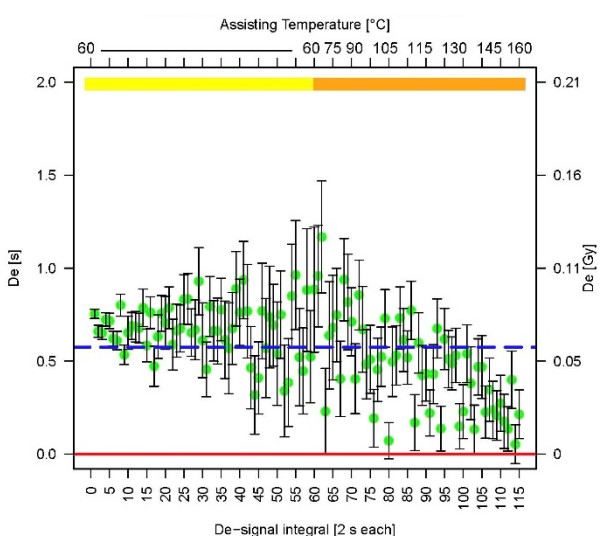

**(a)** PHT 200 °C, $PET_{max}$ and HBL 160 °C

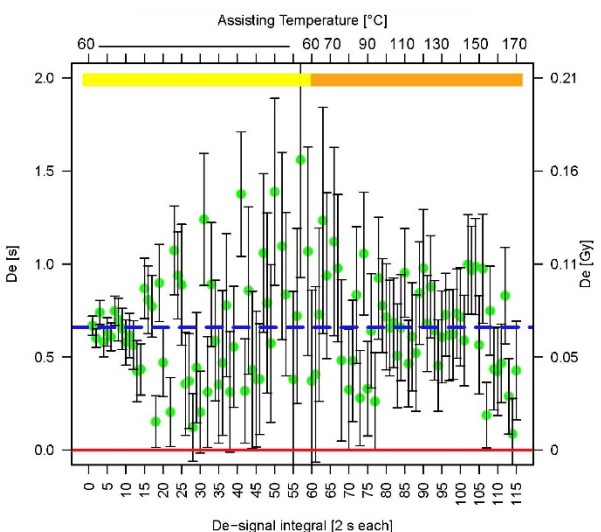

**(b)** PHT 200 °C, $PET_{max}$ and HBL 170 °C

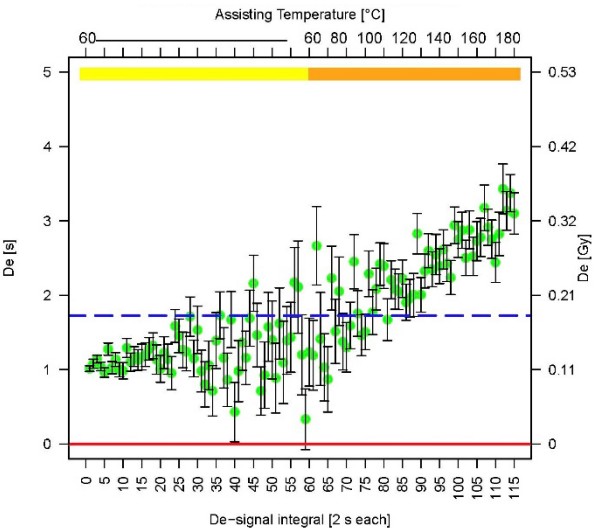

**(c)** PHT 200 °C, $PET_{max}$ and HBL 180 °C

**Figure 8.** Zero-LAB DRTs on the dune sample HDS-1849 with the PET-IRSL SAR protocol with: IR 60 mW cm⁻², 100 s/10.6 Gy dose-response curve, NRM 10 % (1.1 Gy), BLSL hotbleach in each half-cycle of a SAR cycle. PHT 200 °C, while $PET_{max}$ and HBL increase in steps of $\Delta$ 10 K from 160 °C to 180 °C.

For further tests on sample HDS-1849 on variations of the number of hotbleaches or the HBL mode (IRSL instead of BLS or GRSL) and on experiments applying the PET-IRSL step with increased stimulation power density (270 mW cm⁻²



instead of 60 mW cm$^{-2}$), though while using mostly the "hot" SAR parameter values, see **Supplement 3, Fig. S3.4–S3.7,**
**S3.10a–d, S3.11).**

## 5   a-value and g-value determination – Testing the PET-IRSL SAR approach and results

### 5.1   a-value estimation

Whereas beta- and gamma-radiation have comparable luminescence efficiency, the effect of alpha-radiation
represents only a fraction due to a lower penetration depth in matter (relevant for coarse-grains) and heterogeneous energy
deposition concentrated along the alpha tracks (e.g., Zimmerman, 1972). The proportion of alpha to beta efficiency (beta-
radiation was used for De determination in the present study) has to be considered for effective dose-rate and age calculation
and is usually expressed by the so-called a-value (e.g., Aitken, 1985a). If one does not want to use values from the literature,
e.g., for IRSL (Kadereit et al., 2010), pIR$_{50}$IR$_{225}$ (Kreutzer et al., 2014), pIR$_{50}$IR$_{290}$ (Schmidt et al., 2018), a-values have to be
determined in the laboratory, and they should be quantified for a new approach like PET-IRSL SAR. As the typical alpha-
particles may penetrate feldspar only up to ca 20 μm, we ground a limited amount of the 125–212 μm feldspar extracts either
in a ball mill (cf. **section 2.2**) if sufficient material was available (HDS-1849) or manually with pestle and mortar if material
was extremely limited (HDS-1776) and pipetted the fraction 4–11 μm onto FI sample carriers (**section 2.3**). No feldspar
extracts were available for HDS-1827, and only six aliquots of HDS-1776 could be prepared. Aliquots were bleached for one
hour in the respective lexsyg reader (HDS-1776 in LR03, "Pulse"; HDS-1849 in LR04, "Colour"). Afterwards, they received
an alpha-dose (LAB$_{alpha}$) under vacuum in the ELSEC six-seater (cf. **section 2.2**) before being measured with the same PET-
IRSL SAR protocol as used for De measurements.

We performed two measurements on HDS-1849, six aliquots each, one with LAB$_{alpha}$ 146 Gy, the other with LAB$_{alpha}$
89 Gy. Further, we performed a DRT with LAB 100 s/10.6 Gy and a zero-dose test to investigate whether the fine grains
exhibit an unwanted residual dose that might erroneously increase the a-value as observed by Kreutzer et al. (2014). Although
the zero-dose test delivered a slightly elevated background, like coarse grains measured with HBL 180 °C (cf. **Fig. 8c**), this is
negligible with respect to the size of LAB$_{alpha}$ as also corroborated by the DRT (for details cf. **Supplement 5, Fig. S5.2.1–**
**S5.2.4**).


One aliquot of HDS-1776 received LAB$_{alpha}$ 1563 Gy. Two aliquots used for a zero-dose test showed residual doses,
though around 10 % of the beta-dose equivalent of the alpha dose (ca 5–7 % in the low-temperature PET-IRSL range and ca
10–11 % in the very late high-temperature PET-IRSL range of De-signal integral 105–115; cf. **Supplement 5, Fig. S5.1.1–**
**5.1.2**), which appears acceptable. Still, we improved the procedure using the two aliquots that had undergone the zero-dose
test and therefore showed no measurable residual dose. Like the first alpha-dosed aliquot, the two aliquots were administered





LAB$_{alpha}$ 1563 Gy. For data analysis, the alpha-dose was treated like the first half-cycle of a first SAR cycle, and Ln$_{alpha}$/Tn replaced Ln/Tn in the dose-response curve of the preceding zero-dose test.

The results of one representative aliquot of each sample are illustrated in **Fig. 9** (for further aliquots, cf. **Supplement 5, Fig. S5.1.2, S5.2.1, S5.2.3–S5.2.4**). We obtained a-values around ca 0.13–0.14 in the early low-temperature and in the high-temperature PET-IRSL range for HDS-1849, and ca 0.10 in the low-temperature PET-IRSL range and up to ca 0.12 in the high-temperature PET-IRSL range for HDS-1776. The values are in the usual range of a-values applied in luminescence dating (e.g., Faershtein et al., 2020).

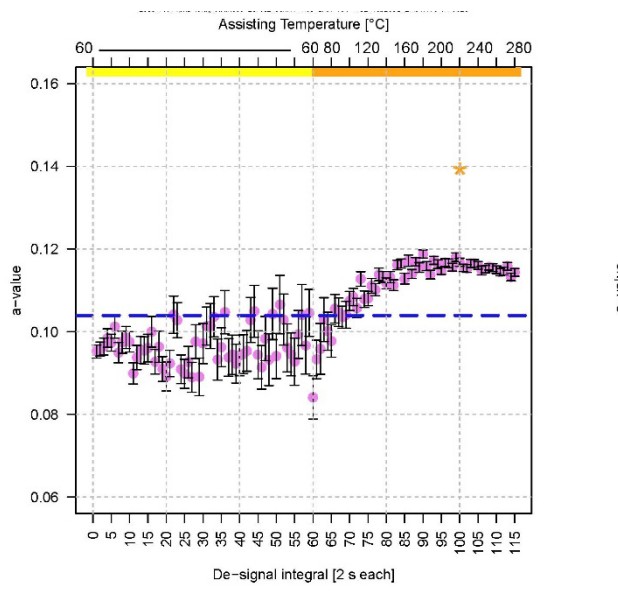

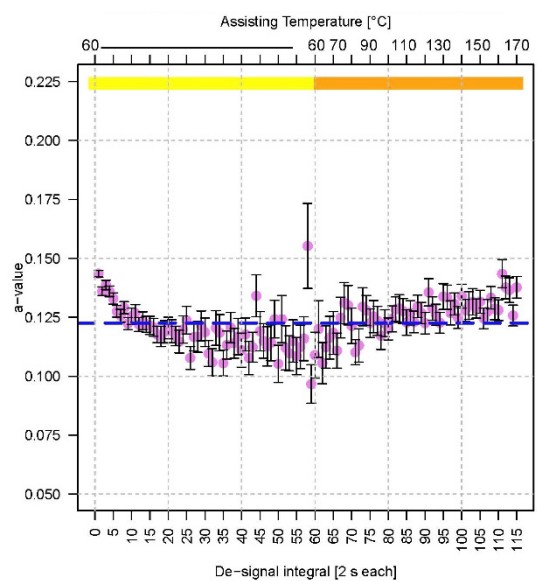

**(a)** HDS-1776 – FG – LAB$_{Alpha}$ 1563 Gy – Aliquot 3     **(b)** HDS-1849 – FG – LAB$_{alpha}$ 89 Gy – Aliquot 5

**Figure 9.** a-value determination on a representative aliquot from **(a)** the loess sample HDS-1776 from the LPS Baix and **(b)** the dune sample HDS-1849 from the Nebraska Sand Hills. For the results of additional aliquots, cf. **Supplement 5, Fig. S5.1.2, S5.2.1, S5.2.3–S5.2.4**). Measurement parameters: IR 60 mW cm⁻², GRSL (HDS-1776) or BLSL (HDS-1849) hotbleach in each subset of a SAR cycle. Same dose-response curves as used for De determination aiming at 2500 s/300 Gy (HDS-1776) with NRM 20 % (60 Gy) and 100 s/10.6 Gy (HDS-1849) with NRM 10 % (1.1 Gy). PHT 320 °C (HDS-1776) and 200 °C (HDS-1849). PET$_{max}$ and HBL 280 °C (HDS-1776), 170 °C (HDS-1849). LAB$_{alpha}$ ca 1563 Gy (HDS-1776) and ca 89 Gy (HDS-1849). Dashed blue line: Mean of all a-values from De integral 1–115 (2 s each). Orange bar: High-temperature PET-IRSL-range (60–280 °C for HDS-1776; 60–170 °C for HDS-1849).



## 5.2   g-value assessment

The luminescence decay, believed to be caused by anomalous fading in feldspar, follows largely linearly the logarithm of time elapsed since irradiation, both for natural and laboratory administered doses (e.g., Huntley and Lamothe, 2001). The rate with reference to the logarithmic time-axis is expressed as g-value as percent signal decline per decade, normalized to two days after laboratory irradiation. The sample was preheated immediately after irradiation, followed by delayed IRSL readout (Auclair et al., 2003). For the fading analysis, we used previously measured and therefore somewhat desensitized aliquots, and they were measured with the same parameter values as used for De determinations. The loess sample HDS-1776 received a laboratory dose (LAB$_{fad}$) of 1750 s/210 Gy, the fluvial sample HDS-1827 of 400 s/48 Gy and the dune sample HDS-1849 of 100 s/10.6 Gy. Measurements started with five immediate IRSL readouts with a ca 1143 s delay after half of the beta-irradiation time of 1750 s for HDS-1776, a ca 440 s delay for HDS-1827, and a ca 288 s delay for HDS-1849, followed by ten readouts after pauses of ca 4 h to 168 h (7 days) and, finally, again three immediate readouts (cf. **Supplement 5, Fig. S5.3.3, Fig. S5.4.3, Fig. S5.5.3**). Mean g-values from the five aliquots of a sample are shown in **Fig. 10** (for g-values as determined for the individual aliquots cf. **Supplement 5, Fig. S5.3.1–S5.5.2)**.

All g-values are in the range ca 0.5–4 % per decade, indicating signal loss. They are above the mean values (cf. dashed blue line) in the early part of the low-temperature PET-IRSL range and below the mean in the (later part of the) high-temperature PET-IRSL range. Disregarding the scatter of the data of HDS-1776 in the low-temperature PET-IRSL beyond De integral 5, the highest g-values are observed for the earliest De-signal integrals. These maxima are around ca 2.2 % per decade for HDS-1776, ca 3.0 % per decade for HDS-1827 and ca 4.1 % per decade for HDS-1849. The dune sample HDS-1849 yields the largest g-values with respect to the mean (cf. dashed blue line) and the earliest values of the low-temperature PET-IRSL. In the high-temperature PET-IRSL range, g-values of HDS-1849 form a plateau around ca 1.8 % per decade, which is also a comparably high value. For HDS-1827, g-values in the high-temperature PET-IRSL range also form a plateau but on a lower level around ca 1.4 % per decade beyond De-signal integral 70, while the last five De-signal integrals (160–170 °C) might show a minimum around ca 0.85 % per decade. For HDS-1776, which was preheated at 320 °C and read out up to PET$_{max}$ 280 °C, g-values drop steadily in the range De-integral 70 to 115 (ca 100–280 °C) from ca 2.0 % per decade to ca 0.9 % per decade. The g-value curves of the individual aliquots (cf. **Supplement 5, Fig. SS5.3.1–S5.5.2)** reveal a more diversified picture behind the mean g-value curves derived from the common analysis of all five aliquots per sample.





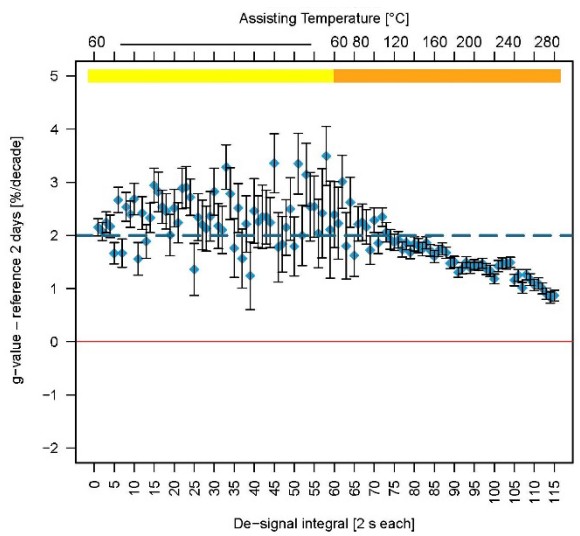

**(a)** HDS-1776

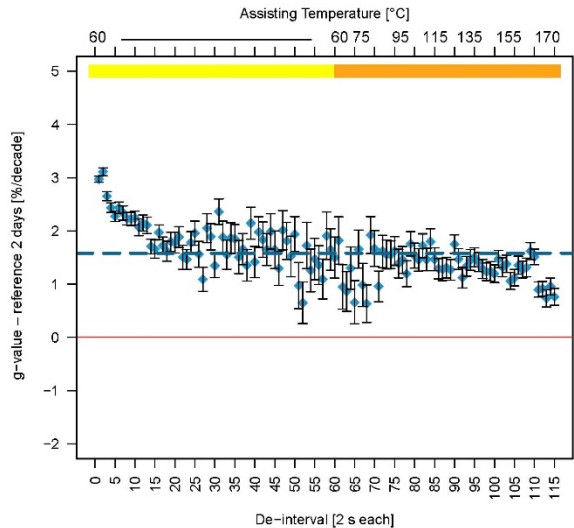

**(b)** HDS-1827

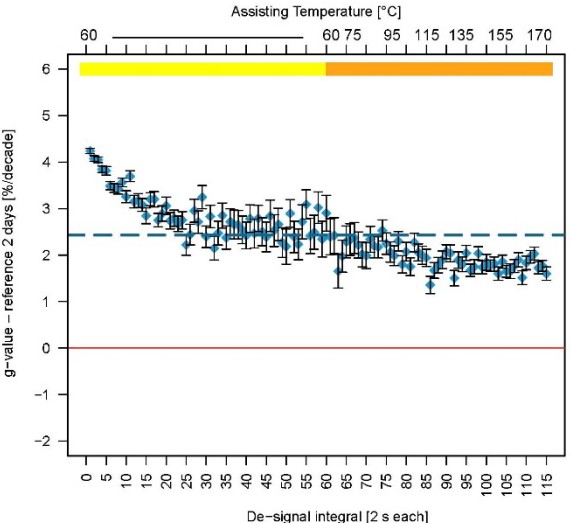

**(c)** HDS-1849

**Figure 10.** g-value determination on the three sediment samples HDS-1776, HDS-1827 and HDS-1849. The fading rates were determined on five aliquots for each sample and represent mean g-values of a sample (for results of the individual aliquots cf. **Supplement 5, Fig. S5.3.1–S5.5.3**). Measurement parameters: IR 60 mW cm⁻², GRSL (HDS-1776, HDS-1827) or BLSL (HDS-1849) hotbleach in each SAR half-cycle. Same dose-response curves as used for De determination aiming at 2500 s/300 Gy (HDS-1776) with NRM 20 % (60 Gy), 1000 s/120 Gy (HDS-1827) with NRM 10 % (12 Gy) and 100 s/10.6 Gy (HDS-1849) with 10 % (1.1 Gy) LAB_fad 1750 s/210 Gy (HDS-1776), 400 s/48 Gy (HDS-1827) and 100 s/10.6 Gy (HDS-1849). PHT 320 °C (HDS-1776), 200 °C (HDS-1827, HDS-1849). PET_max and HBL 280 °C (HDS-1776), 170 °C (HDS-1827, HDS-





1849). Five immediate readouts with ca 1143 s delay after half of the beta irradiation time of 1750 s (HDS-1776), ca 440 s delay after half of the beta irradiation time of 400 s (HDS-1827) and ca 288 s delay after half of the beta irradiation time of 100 s (HDS-1849) followed by ten readouts after pauses of ca 4 h to 168 h (7 days) and, finally, three immediate readouts. Dashed blue line: Mean of all g-values from interval 1–115 (2 s each). Thin red line: Potential level of zero fading. Yellow bar: Low-temperature PET-IRSL range (constant at 60 °C). Orange bar: High-temperature PET-IRSL-range (60–280 °C for HDS-1776; 60–170 °C for HDS-1827 and HDS-1849).

## 6 Further experiments and results

### 6.1 PET-IRSL spectra

To assess whether the blue feldspar emission is suitable for testing the PET-IRSL SAR approach on these samples, we recorded IR stimulated emission spectra covering the range ca 350–650 nm (UV to red; resolution of ca 0.5 nm). A background spectrum obtained at room temperature from an empty sample holder was subtracted from the measurements prior to any data processing. To account for the wavelength-dependent transmission of the filter, diffraction grating, CCD camera, and fiber optic bundle, spectra were efficiency-corrected using a spectral-response function obtained from the product of efficiency curves provided by the manufacturers (see Sontag-González et al., 2022).

We measured the PET-IRSL spectrum of a previously measured aliquot of the Nebraska dune sample HDS-1849 up to PET$_{max}$ 280 °C after PHT 320 °C (**Fig. 11).** It exhibits a clearly defined blue emission (**Fig. 11a**) which, like the PET-IRSL signal curves (e.g., **Fig. 3**), starts with a prominent IRSL shine-down and, after reaching a background level, rises to form a broad though well-defined PET-peak in the high-temperature PET-IRSL range (here ca 150–180 s, ca 160–220 °C). A yellow emission is hardly present after PHT 320 °C (**Fig. 11b**; for IRSL spectra of all three sediment samples see **Supplement 6, Fig. S6.7.1 –S6.7.6**).



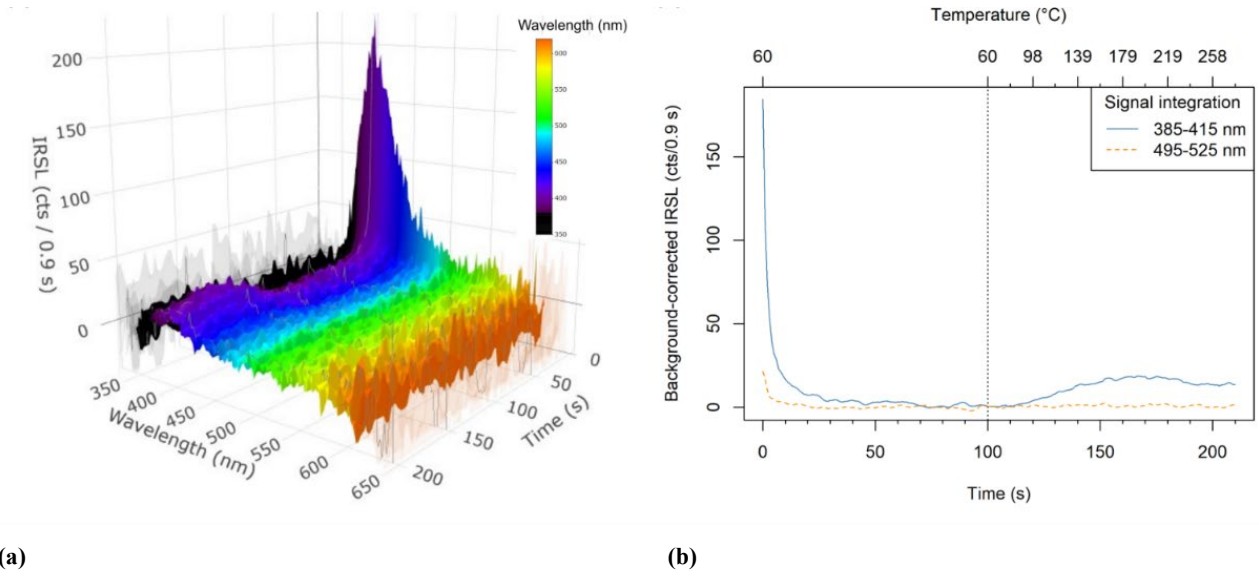

**(a)**                                                    **(b)**

**Figure 11.** PET-IRSL spectrum of sample HDS-1849. The aliquot was dosed to ~400 Gy and preheated at 320 °C for 60 s. PET-IRSL readout occurred for 100 s at 60 °C (instead of 120 s low-temperature PET-IRSL readout in most of the other measurements) before the temperature was raised (2° C/s) to 280 °C. Panel **(b)** shows the decay curves of the same data for two integration windows targeting possible blue and yellow emissions.

## 6.2 Testing the effect of different bleaching intensities on the PET-IRSL De-value curve

Sample HDS-1776 was used to investigate whether PET-IRSL may indicate partial bleaching. For this test, we imagined a sequence of two events of reworking of the same sediment material, a first reworking (penultimate event) in which
the latent luminescence signal was sufficiently reset and subsequently – after an imagined interim sediment storage – a second reworking (last event) with partial bleaching. The scenario was executed in the lexsyg reader: **(1)** The aliquot was bleached for one hour under the solar simulator, simulating thorough bleaching similar to the DRTs (**section 4.1**), but here representing not the last event of sediment reworking, but the penultimate event. **(2)** The aliquot received a dose of 1250 s/150 Gy to simulate storage in the sediment archive. **(3)** TL preheating at 320 °C for 60 s accomplished the simulation of long-time
storage. **(4)** The sample was partially bleached in the solar simulator using varied light exposure times of ≤ 1 h. Experiments were performed with white light as well as monochromatic light where the red, yellow, green, blue and UV bleaches were administered at the same energy level as the bleaches performed using white light (cf. bleaching for DRTs in **section 4.1**). This scenario was designed to simulate partial bleaching during the sample's last exposure to sunlight prior to its final burial (last event). Bleaching with red, yellow or green light may imitate bleaching during sunset or in (turbid) water. **(5)** The sample
received an additional dose of 1250 s/150 Gy to simulate another period of sediment storage after burial. **(6)** The sample was analyzed with the PET-IRSL SAR protocol using the same parameter values as those chosen for DRT or De determination, starting with a TL preheating at 320 °C for 60 s. Thus, a recovered dose of ca 1250 s/150 Gy would point to the last bleaching





and burial event, while a potentially recovered dose of ca 2500 s/300 Gy (summed partial doses, 1250 s/150 Gy each) would indicate the penultimate event.


Aliquots were repeatedly used, which may explain the fluctuant course of the De-value curves of some tests (e.g., after yellow light bleaching, **Supplement 1, Fig. S1.3.3**). We repeated selected PBL tests on previously unmeasured aliquots. Taking inter-aliquot heterogeneity into account, the basic features observed on the repeatedly used aliquots were reproducible (cf. **Supplement 1, Fig. S1.2.1–1.4.2**).


Results of selected PBL tests in **Fig. 12** and **Fig. 13** show De-value curves that differ from those of the DRTs on well-bleached aliquots, which, when measured with an appropriate thermal treatment (PHT, $PET_{max}$, HBL) and NRM, exhibit long De-value plateaus (**Fig. 5, Fig. 6c–6d**). The graphs compiled in **Fig. 12** and **Fig. 13** show that not only does the mean De value (dashed blue line) decrease with increasing bleaching intensity during the last event, but also illustrate a sequence of changes

in the shape of a De-value curve.

The initial part of the low-temperature PET-IRSL range reacts to the slightest partial bleaching by either very short-lasting white light (**Fig. 12a**) or longer lasting long-wavelength light stimulation by a downward bend of the De-value curve (**Fig. 13a**). For red-light bleaching an initial hook-shape is present and remains after 960 s partial bleaching (cf. **Supplement 1,**

**Fig. S1.3.1d**), while at the same time, the De values in the late high-temperature PET-IRSL form an upper plateau at approximately the level of the sum of the two administered doses (penultimate event; 2500 s/300 Gy). The combination of an initial hook-shape and an upper plateau in the high-temperature PET-IRSL range may serve as a proxy of very modest bleaching. PBL tests with 1 s and 3 s partial bleaching with white light on HDS-1827 and HDS-1849 corroborated this assumption as these tests, too, showed an initial hook-shape and an upper plateau around the value of the two summed partial

doses (**Supplement 2, Fig. S2.15; Supplement 3, Fig. S3.21**).

The upper plateau of De values shortens (lateral shift of the plateau edge from left to right, in the direction of increasing temperature) with increasing bleaching intensity as illustrated for 1 s white-light bleaching (**Fig. 12a**) or 60 s yellow-light bleaching (**Fig. 13b**). Further increasing bleaching intensity leads to further erosion of the upper plateau by rounding the

plateau edge as observed for 3600 s red-light bleaching (**Fig. 13c**) and lowering it as illustrated for 960 s green-light bleaching (**Fig. 13d**). Further increase in bleaching intensity leads to the complete erosion of the upper-level plateau leaving a slope of De values rising from a low-temperature De-value plateau. The gradient of the De-value slope decreases with increasing bleaching intensity, as illustrated by 10 s and 120 s white-light bleaching (**Fig. 12b, 12c**) and – for monochromatic bleaching – 60 s blue-light bleaching and 60 s UV bleaching (**Fig. 13e, 13f**). The final stage in the sequence of increasing bleaching

intensity is an end-to-end plateau on the lower level, as illustrated by 960 s white-light bleaching (**Fig. 12d**). The mean of all



De values of 1331.2 s/160 Gy recovers the last event, associated with a dose of 1250 s/150 Gy, within a tolerable error margin of 10 %.

The PET-IRSL SAR results of the monochromatic PBL tests show a decreasing bleaching intensity from UV stimulation to blue-light, green-light, yellow-light and red-light stimulation in agreement with observations from previous monochromatic bleaching tests and tests with filtered light (e.g., Frouin et al., 2015; Kars et al., 2014).

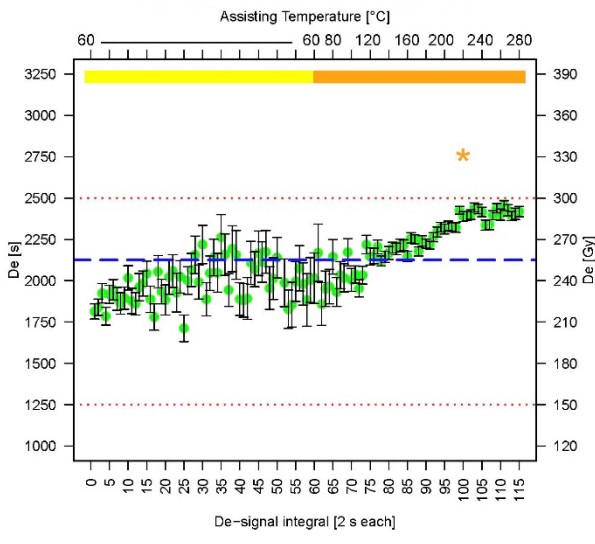

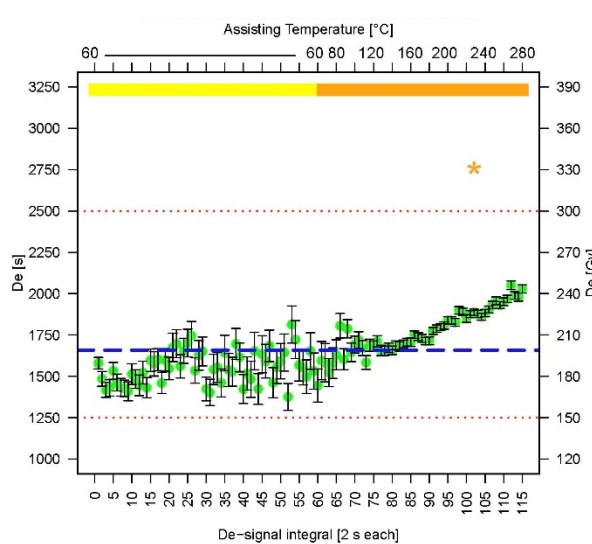

**(a)** HDS-1776, PBL white, 1 s

**(b)** HDS-1776, PBL white, 10 s

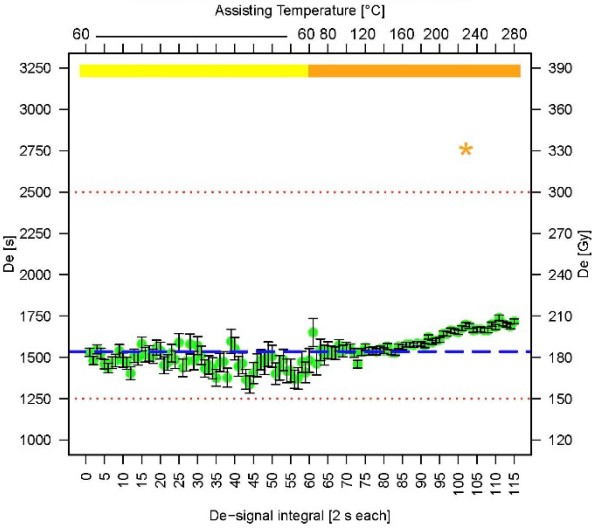

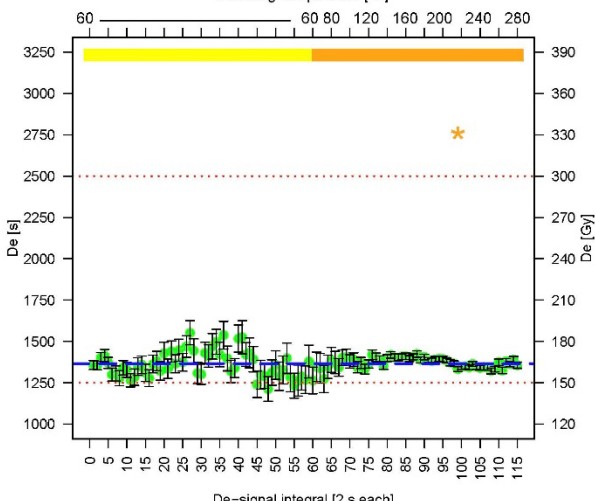

**(c)** HDS-1776, PBL white, 120 s

**(d)** HDS-1776, PBL white, 960 s





**Figure 12.** Selected results of PBL tests with white light on repeatedly used aliquots of the loess sample HDS-1776 from the LPS Baix in southern France (for results of further PBL tests both on repeatedly used aliquots and on unused aliquots cf. **Supplement 1, Fig. S1.2.1–1.2.2, S1.4.1**). Measurement parameters: IR 60 mW cm⁻², GRSL (HDS-1776) hotbleach in each SAR half-cycle. Same dose response curves as used for De determination aiming for HDS-1776 at 2500 s/300 Gy with NRM 20 % (60 Gy). PHT 320 °C, PET$_{max}$ and HBL 280 °C. Dashed blue line: Mean of all De-values from signal integral 1–115 (2 s each). Thin red dotted lines: Administered partial doses, 1250 s/150 Gy after initial thorough signal resetting (penultimate event) plus 1250 s/150 Gy after the partial bleaching (last event). The summed dose associated with the penultimate event is indicated by the upper red dotted line at 2500 s/300 Gy. 1250 s/150 Gy dose associated with the last event indicated by the lower red dotted line at 1250 s/150 Gy. Yellow bar: Low-temperature PET-IRSL range (constant at 60 °C). Orange bar: High-temperature PET-IRSL-range (60–280 °C).

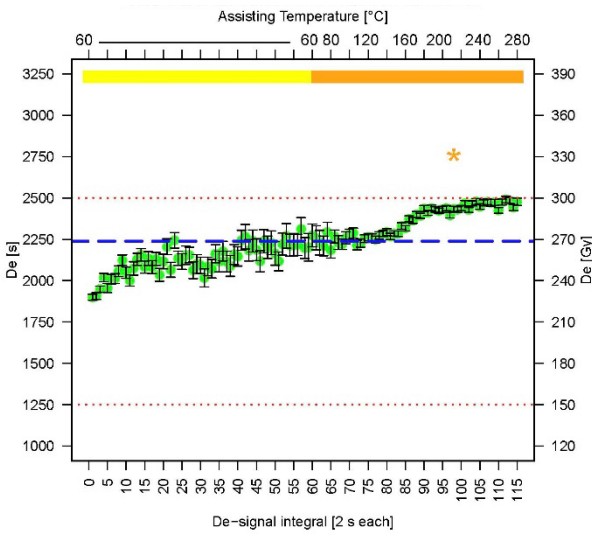

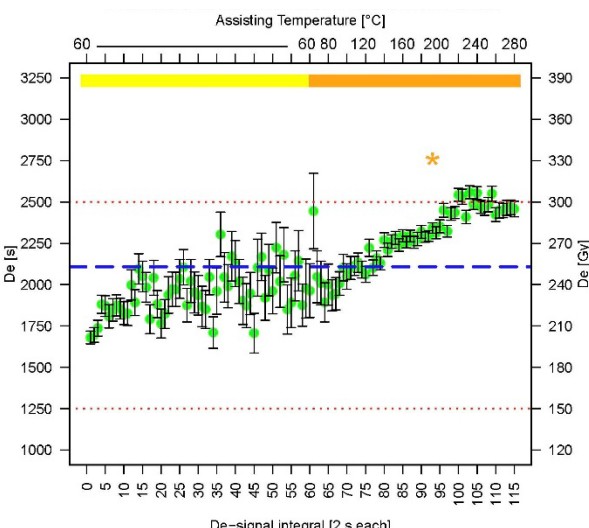

**(a)** HDS-1776, PBL red, light 15 s           **(b)** HDS-1776, PBL yellow, light 60 s, unmeasured





**(c)** HDS-1776, PBL red, light 3600 s

**(d)** HDS-1776, PBL green, light 960 s

**(e)** HDS-1776, PBL blue, light 60 s

**(f)** HDS-1776, PBL UV, light 60 s

**Figure 13.** Selected results of PBL tests with monochromatic light bleaching. Except (b), previously measured (De determination, DRT) and for PBL tests repeatedly used aliquots of the loess sample HDS-1776 from the LPS Baix in southern France (for results of further PBL tests cf. **Supplement 1, Fig. S1.3.1–1.3.10, S1.4.2**). Measurement parameters: IR 60 mW cm$^{-2}$, GRSL (HDS-1776) hotbleach in each SAR half-cycle. Same dose-response curves as used for De determination aiming for HDS-1776 at 2500 s/300 Gy with NRM 20 % (60 Gy). PHT 320 °C, PET$_{max}$ and HBL 280 °C. Dashed blue line: Mean of all De-values from signal integral 1–115 (2 s each). Thin red dotted lines: Administered partial doses, 1250 s/150 Gy after initial thorough signal resetting (penultimate event) plus 1250 s/150 Gy after the partial bleaching (last event). The summed dose associated with the penultimate event is indicated



by the upper red dotted line at 2500 s/300 Gy. 1250 s/150 Gy dose associated with the last event indicated by the lower red dotted line at 1250 s/150 Gy. Yellow bar: Low-temperature PET-IRSL range (constant at 60 °C). Orange bar: High-temperature PET-IRSL-range (60–280 °C).


## 6.3 Effect of different preheat temperatures on the shape of the PET-IRSL signal curve

The effect of different preheat temperatures on the PET-IRSL signal curve was tested in seven cycles on a previously measured aliquot of HDS-1776, repeatedly administering LAB 2500 s (300 Gy) followed by PHT, PET-IRSL readout and GRSL hotbleach (cf. **Fig. 2,** steps 8 and 1–3). PHT was increased from 200 °C to 320 °C in seven steps of 20 K. PET-IRSL

readout occurred in each cycle up to 280 °C. As, however, PET-IRSL readout is not useful beyond the preheat temperature, for graphical presentation, the recorded data beyond the preheat temperature was not shown (**Fig. 14**, data curves for PHT 300 °C and 320 °C showing the final PET-IRSL decline, while for PHT 200–280 °C ending abruptly). This procedure ensured that the temperature ramp of the PET-IRSL custom curve (step 2 in **Fig. 2**) stayed the same (2 K/s) in each cycle and did not become a variable parameter value in the measurement setup.


All data curves show an initial shine-down in the 60 °C range and a rise towards a PET-peak in the high-temperature PET-IRSL, as known from **Fig. 3**. The peak position shifts with increasing preheat temperature from ca 160 °C readout-temperature for PHT 200 °C to ca 220 °C for PHT 320 °C, while the signal (peak height) decreases by ca 45 %. In contrast, the signal strength at the start of the 60 °C shine-down reduces to ca 4.5 %, i.e., by a factor of roughly 10. While the low-

temperature signal is much higher than the high-temperature signal after a low preheat temperature, the ratio changes after higher temperatures, especially the highest preheat temperature of 320 °C. As the high-temperature signal after preheat 280 °C is ca 160 % of that after preheat 320 °C, for samples with low signal strength, it might be advantageous to reduce the preheat temperature below 320 °C.





**Figure 14.** PET-IRSL signal curves of sample HDS-1776 after different preheat temperatures (PHT). **(a)** Preheat in the range 200–320 °C. **(b)** Preheat in the range 260–320 °C.

## 6.4 Testing whether PET-readout leads to recuperated TL

Preheating prior to PET-IRSL readout depletes the corresponding TL signal up to the preheat temperature. However, for pIRIR (Wang et al., 2014) suggested that electrons stimulated by IR$_{1st}$ are recaptured in TL traps and read out again by IR$_{2nd}$, leading to electron recycling. Therefore, we investigated whether the PET-peak represents a TL-peak resulting from electrons redistributed into a TL-trap by low-temperature IRSL. Similar to our test investigating the effects of differing preheat temperatures (**section 6.3**) we repeatedly applied an abbreviated PET-IRSL cycle with a beta dose of 2500 s (300 Gy) to HDS-1776, 400 s (47 Gy) to HDS-1827 and 100 s (10.6 Gy) to HDS-1849, like in the previous test skipping the normalization-dose half-cycle, utilizing previously measured aliquots. PHT was, as in the De measurements (**section 7.2**), 320 °C for HDS-1776, and 200 °C for HDS-1827 and HDS-1849. In contrast to ordinary PET-IRSL SAR measurements, IR-stimulation was switched off early after varying duration, while thermal assistance (now pure TL) was recorded up to virtual PET$_{max}$





(280 °C for HDS-1776; 170 °C for HDS-1727 and HDS-1849). At the beginning and at the end of a complete measurement, a PET-IRSL readout with IR-stimulation up to $PET_{max}$ was recorded (red and dark red lines in **Fig. 15**), monitoring the sensitivity change from the first to the last cycle. Between these two completed PET-IRSL cycles, we measured further cycles with IR-stimulation switched off after 60 s (green line), corresponding to the first half of the 60 °C range, and after 120 s (black line), corresponding to the complete 60 °C range, but further recorded the signal produced by the thermal assistance (TL) up to

$PET_{max}$. This way, we could monitor potential TL-signal recuperation. For the high-temperature range, we extended the IR-duration in steps of 10 s corresponding to $\Delta$ 20 K for HDS-1776 (80–270 °C; 10 steps, 20 K each) and $\Delta$ 10 K for HDS-1827 and HDS-1849 (70–160 °C; 10 steps, 10 K each) while recording the purely thermally assisted (TL) signal up to $PET_{max}$.

All cycles show almost identical signal curves up to the point of IR-switch-off, when each signal declines and reaches

its background (cf. **Fig. 3**). The late background signals show a slight steady increase, similar to – and perhaps identical with – those of the zero-dose cycle in a PET-IRSL SAR measurement (recuperation signal; cf. purple line in **Fig. 3**). The longer the IR-stimulation lasts, the higher the signal rises above the background. For pIRIR, aliquots with an elevated background signal are often eliminated by applying a threshold for the recuperation-rejection criterion, a procedure that could also be included in PET-IRSL.


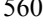

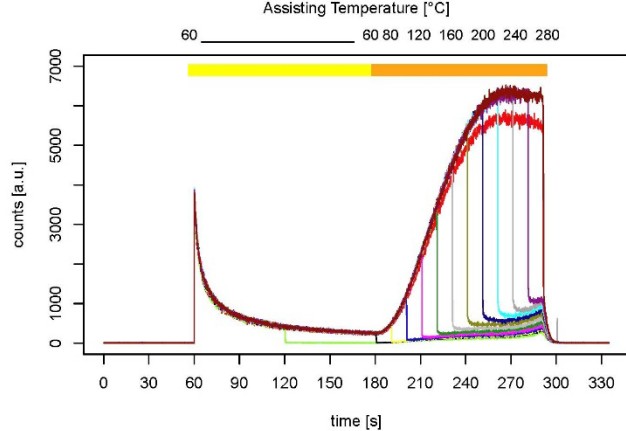

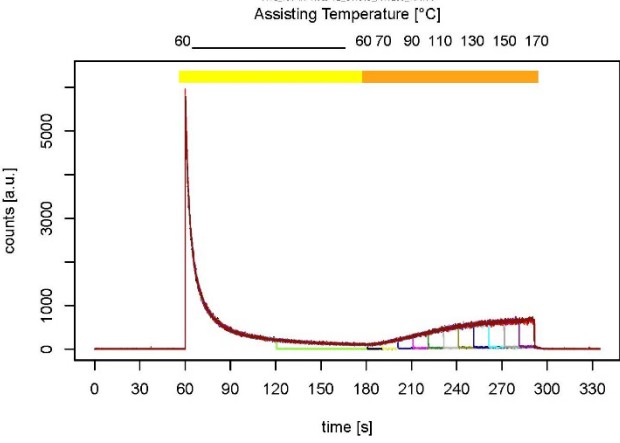

**(a)** HDS-1776 (Baix, loess) – $PET_{max}$ 280 °C

**(b)** HDS-1827 (BSN, river bed deposit) – $PET_{max}$ 170 °C





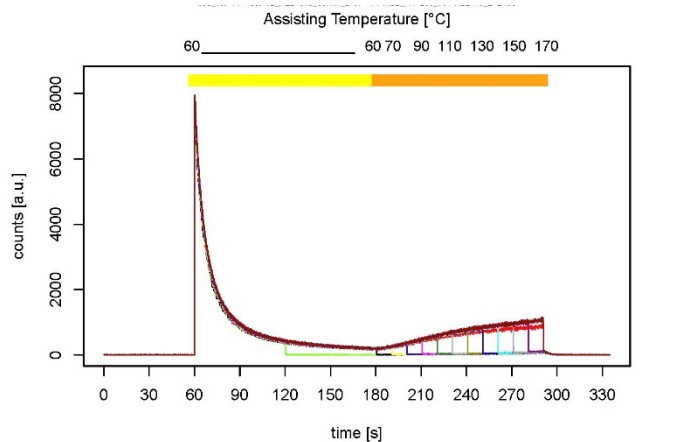

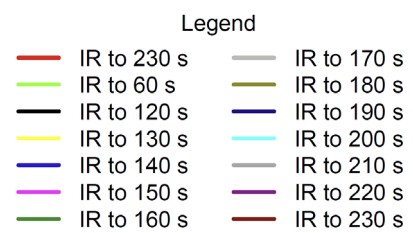

**(c)** HDS-1849 (Nebraska, dune sand) – PET$_{max}$ 170 °C

**Figure 15.** Variable durations of PET-IRSL readout for samples HDS-1776, HDS-1827 and HDS-1849. Aliquots previously used in De measurements. IR-stimulation superimposed on the thermally customized TL curve starting with 230 s (red) to PET$_{max}$ (60–290 s measurement time shown on x-axis); PET$_{max}$ equal **(a)** 280 °C for HDS-1776 and **(b, c)** 170 °C for HDS-1827 and HDS-1849). Followed by 60 s (green) and 120 s (black) IR-stimulation both at 60 °C (60–180 s); then extending IR-stimulation with increasing assisting temperature in steps of 10 s to 130 s (yellow), 140 s (blue), 150 s (magenta), 160 s (dark green), 170 s (gray), 180 s (olive), 190 s (darkblue), 200 s (cyan), 210 s (dark grey), 220 s (purple) and – again – 230 s (dark red). 10 s PET-IRSL readout-time corresponding to **(a)** Δ 20 K for HDS-1776 and **(b, c)** Δ 10 K for HDS-1827 and HDS-1849. No signal normalization, therefore, the two 230 s curves (red vs dark red; first vs last cycle) may deviate from one another (sensitivity change). Almost no difference for **(b)** HDS-1827. Strongest difference for **(a)** HDS-1776 with stronger PHT (320 °C vs 200 °C) and higher PET$_{max}$ (280 °C vs 200 °C)).

# 7   Discussion

## 7.1   Discussion of the method

Despite differing geographic origins, palaeoenvironments, palaeodoses and preparation, the sediment samples, as well
as the commercial feldspar specimens, showed similar features under PET-IRSL treatment. IR-stimulated spectra revealed a clearly defined blue emission (ca 410 nm; **Fig. 11**; **Supplement 6, Fig. S6.7.x–S6.7.y**), and all discussed PET-IRSL features relate to this blue feldspar emission. Like pIRIR and MET, PET-IRSL separates less stable from more stable IRSL signals chronologically by sequential IRSL-readout at increasing temperatures but delivers a quasi-continuous data record.

To clear out the less stable low-temperature PET-IRSL signal, initial PET-IRSL at 60 °C lasted for 120 s, before temperature-ramping with IR-stimulation up to PET$_{max}$ occurred with 2 K/s for PET$_{max}$ 280 °C and 1 K/s for PET$_{max}$ 170 °C. This leads to a bisected PET-IRSL signal curve with a low-temperature PET-IRSL range showing a shine-down as known





from IRSL, followed by a signal rise at high-temperature PET-IRSL, which at ca 200–220 °C forms a dome-shaped PET-peak. Likewise, the De-value curves show a low- and high-temperature PET-IRSL range, which may perform differently in test and
De measurements.

### 7.1.1 PET-IRSL signal curve

The general shape of a PET-IRSL signal curve shows minor variations depending on sample and/or SAR parameters,
especially thermal (pre-)treatment; however, the general pattern is remarkably similar for all investigated samples.

*Ratio of signal counts of the onset of the initial shine-down to the PET-peak*

For De measurements and NRM tests of HDS-1776 with PHT 320 °C the luminescence signals of the onset of the
shine-down and the PET-peak may be roughly equally high (e.g., **Supplement 1, Fig. S1.6.1–S1.6.2**), but in most cases the PET-peak is higher (e.g., **Supplement 1, Fig. S1.6.3–S1.6.4**), as illustrated in **Fig. 3**. For aliquots subjected to PBL tests (cf. **section 6.2**) the PET-peak may be higher, commonly by a factor > 2 for UV-bleaching (e.g., **Supplement 1, Fig. S1.6.7**), likely reflecting better bleachability of the early PET-IRSL signal. Potentially, a decreased ratio of the size of the early low-temperature PET-IRSL signal versus that of the high-temperature PET-peak of a natural palaeodose may be caused by higher
fading of the first and reduced fading of the latter. PET-IRSL may provide g-value curves (cf. **section 5.2**) to help clarify this issue.

Stronger preheat reduces the low-temperature PET-IRSL signal disproportionally stronger than the high-temperature PET-IRSL signal to the point of an inversion of the ratio of the two (cf. **Fig. 14**).

For Norfloat-Potash-Feldspar (12.0 wt.% $KO_2$), G-40-Feldspar (10.4 wt.% $KO_2$) and F-20-Feldspar (4.1 wt.% $KO_2$), the height of the PET-peak with PHT 320 °C represents only a fraction of the initial values of the shine-down (cf. **Supplement 4, Fig. S4.1.1, S4.2.1, S4.3.1**). Differences in the low-temperature to a high-temperature signal have also been observed for $pIR_{1st}IR_{2nd}$. Buylaert et al. (2009) found that $IR_{220}$ signals are 1.6 times higher compared to $IR_{50}$; Firla et al. (2024)
observed both, relatively smaller and relatively larger $IR_{50}$ signals compared to $IR_{225}$ signals.





*PET-peak temperature*

The position of the PET-peak varies between feldspar specimens but is rather constant for repeat measurements of the same aliquot. PET-peak positions are at De-interval 92 (ca 188 °C; F-20), De-interval 96–98 (ca 204–212 °C; G-40), and De-interval 100 (ca 220 °C; Norfloat-Potash), shifting to higher readout temperature with increasing potassium content. Likewise, the slight shifts of peak positions observed for HDS-1776 after PHT 320 °C (**Supplement 1, Fig. S1.6.5–S1.6.6,** De-signal integral 97 or 208 °C in **Fig. S1.6.5**; De-signal integral 102 or 228 °C in **Fig. S1.6.6**), might indicate varying feldspar-types/potassium-contents dominating the luminescence signal of an aliquot. Whether the shifts constitute causality between 610 peak position and potassium content cannot be determined based on the limited number of investigated specimens.

Tests on sample HDS-1776 with an unchanged temperature ramp of 2 K showed that the PET-peak shifts systematically to slightly earlier readout-times with decreasing preheat temperatures (**Fig. 14**).

*Shape of the PET-peak*

Among the three commercial feldspar specimens, the most pronounced PET-peak is exhibited by Norfloat-Potash with the highest potassium content. The least pronounced and broadest PET-peak is observed for G-40, while the specimen with the lowest potassium content (F-20) shows a slightly more developed PET-peak. We tentatively conclude that potassium 620 content does not determine the variations in the shape of the PET-peak.

When temperature ramping in the PET-IRSL custom curve is gradually decreased after decreasing PHT (down to 1 K/s for $PET_{max}$ 170 °C after PHT 200 °C), a PET-peak is observed for PHT ≥ 260 °C. The apex becomes less clear for lower preheat temperatures until no peak position can be identified (**Supplement 2, Fig. S2.17–S2.18**; **Supplement 3, Fig. S3.23**).

To preclude that the PET-peak results from an underlying TL signal rather than the recombination of electrons beyond a threshold of thermally progressively assisted IR-stimulation, control measurements were performed (**Fig. 14**). The observed TL signal, between IR switch-off and potential $PET_{max}$, showed some steady rise above the background, resembling the PET-IRSL signal of a SAR-recuperation step, but it did not show the characteristic shape as the PET-peak. Therefore, we conclude 630 that charge recycling for PET-IRSL is small compared to $pIR_{1st}IR_{2nd}$ (Wang et al., 2014). A connection to the different readout techniques seems likely. Whereas with pIRIR, a sample cools down between separate IRSL readouts, progressive increase of thermal assistance in PET-IRSL might prevent (exhaustive) TL recuperation. If a minor fraction of recycled electrons still contributes to the slightly increasing background signal in **Fig. 15**, this could explain g-values above zero in the (late) high-temperature PET-IRSL reach (cf. **Fig. 10**).





### 7.1.2    SAR parameters and De-value curve

For De determination as detailed in **section 2.4**, we did not apply any rejection criteria normally used for pIR$_{1st}$IR$_{2nd}$. This omission seems justified as PET-IRSL SAR delivers a chain of data points, and the course of De values shows whether these are plausible or erratic. Depending on the signal intensity, the uncertainty of an individual De value may vary. Therefore, it is always lowest for the low-temperature PET-IRSL range beyond De-signal integral ca. 20.

To enhance the precision for De determination, a mean of several successive De values can be calculated. Alternatively, the length of De-signal integrals could be increased and/or longer data channels could be employed. Additionally, higher IR-power settings may improve the signal-to-noise ratio and reduce the errors/scatter of the individual De-data points (cf. e.g., **Supplement 3, Fig. S3.24a–S3.24b, Fig. S3.10c, S3.10f**). Finally, larger aliquots can increase signal intensity, at the cost of a mixed signal, though. We kept the IR-power (60 mW cm$^{-2}$; except few tests), the length of the data channels (0.1 s/data channel) and the number of data points (115) the same, but used smaller aliquots for the brighter sample HDS-1776 and larger aliquots for the dimmer samples HDS-1827 and HDS-1849 (cf. **section 2.3**).

*SAR parameters – NRM and PHT*

A SAR measurement represents a complex system with its parameters, including the number and sizes of REGs used for dose-response curve construction, influencing measurement results multifold and repeatedly. The PET-IRSL De-value curve likewise responds to each parameter that is changed in the course of adapting the SAR protocol to a sample.

Our PET-IRSL SAR measurements suggest that even more important than the size of NRM is an appropriate thermal (pre-)treatment. For the samples we analyzed, LAB was recovered correctly if PHT was down-regulated sufficiently. 10 % NRM worked successfully in each case, recovering different LABs reliably as shown by end-to-end De-value plateaus.

When applying PHT 320 °C with PET$_{max}$ and HBL 280 °C, which performed well for HDS-1776 with a 2500 s/300 Gy dose-response curve and LAB 2500 s/300 Gy, also for HDS-1827 and HDS-1849, yet with a 1000 s/120 Gy and a 100 s/10.6 Gy dose-response curve, respectively, testing LAB 1000 s/120 Gy and 400 s/48 Gy (HDS-1827) or 100 s/10.6 Gy and 50 s/5.3 Gy (HDS-1849), no satisfying results were obtained. While the expected dose was mostly (HDS-1849) or sometimes (HDS-1827) met by the high-temperature PET-IRSL around (HDS-1827) or beyond (HDS-1849) the PET-peak (**Fig. 6a, 6b**), the low-temperature PET-IRSL always underestimated LAB.



To determine whether De underestimation for the low-temperature PET-IRSL is specific to PET-IRSL, we applied $pIR_{60}IR_{280}$ NRM tests to HDS-1849 with PHT 320 °C, a 100 s/10.6 Gy dose-response curve and LAB 100 s/10.6 Gy, testing NRM 2.5–100 % (see **Supplement 6, Fig. S6.4.1–S6.4.2**). For $IR_{60}$, the expected De was always underestimated by 40–15 %, thus corroborating the PET-IRSL results. LAB was recovered well by $pIRIR_{280}$ for NRM 30 % and 40 %. One of these would be the favored NRM for De measurements with a $pIR_{60}IR_{280}$ SAR protocol, as they provide the expected De. The irregular shapes of the PET IRSL De-value curves (cf. **Fig. 6a, 6b**) suggest they should not be used for generating SAR data but instead require that further tests be performed until continuous plateaus are achieved.

*Zero-dose tests – PHT – $PET_{max}$ - HBL*

Zero-dose tests proved most important for adapting the thermal SAR parameters as the shape of the De-value curve responds highly sensitively to variations of PHT, $PET_{max}$ and HBL. A flat end-to-end De-value curve – from the beginning of the low-temperature PET-IRSL to the end of the high-temperature PET-IRSL – possibly around zero or close above the zero-line (cf. **Fig. 7b**, **Fig. 8b**) seems a prerequisite to produce good results for samples with (low) natural palaeodoses. Unsuitably high PHT will cause an increase in recovered PET-IRSL De values and an increase to an even higher level for the high-temperature PET-IRSL, thus distinguishing the performances of the low- and high-temperature PET-IRSL (e.g., **Fig. 7a**, see also **Supplement 2, Fig. S2.10a–d**). Once the optimal preheat temperature is known, the temperatures for $PET_{max}$ and HBL must be optimized. For PET-IRSL, the prudent selection of thermal sample (pre-)treatment is crucial, as a rising De-value curve as a consequence of excessive PHT may be misinterpreted as partial bleaching.

Depending on the sample, the De-value curve may form a longer (semi-)plateau along the high-temperature PET-IRSL range (e.g., **Supplement 2, Fig. S2.10a–d**) or it may form a miniature plateau beyond the PET-peak (**Supplement 3, Fig. S3.15–S3.17**). The question of why some samples responded earlier and longer, and others later to the high-temperature PET-IRSL was not in the scope of this study. However, we hypothesize that the observation correlates with different band-tail-states becoming involved with respective temperature assistance of the IR-readout.

Such an upper (longer or miniature) plateau was also observed in malfunctioning DRTs (cf. **Fig. 6a, 6b**; **Supplement 3, Fig. S3.1)** or De measurements (not shown here), if the PHT is chosen inappropriately high for a comparatively small LAB or De. For larger De values/LABs, the high-temperature PET-IRSL rise (see **Fig. 7a**) is not noticeable any longer (cf. **Fig. 5a, Fig. 5b**). The slightly elevated De-value curve at high-temperature PET-IRSL in **Fig. 5b** appears to be rather the result of the twice as large NRM as compared to **Fig. 5a**.

For the loess sample HDS-1776 measured with a 2500 s/300 Gy dose-response curve, a strong PHT 320 °C as well as $PET_{max}$ and HBL 280 °C, a mean residual dose of ca 71 s/8.5 Gy was determined, with individual values scattering between



2.3 s/0.28 Gy and, in one case, 184 s/22.1 Gy in the low-temperature PET-IRSL range and around 100 s/12 Gy in the high-temperature PET-IRSL around the PET-peak (**Fig. 7a**). The mean compares to a fraction of ca 5 %, the fraction around the PET-peak to ca 7 % of the smallest De values around ca 1500 s/180 Gy obtained for the natural dose (**Fig. 16**). The strong
preheat procedure is advantageous with respect to the expected stability of the IRSL signal (Buylaert et al., 2012; Murray et al., 2009). As the protocol parameter values do not increase the recovered residual dose beyond a tolerable limit, they are deemed appropriate for palaeodose assessment at the LPS Baix.

If, instead, a low absolute residual dose is required, e.g., for young samples, the preheat temperature needs to be
reduced. However, lower PHTs come most likely at the cost of the PET-IRSL signal stability (anomalous fading) (cf. Murray et al., 2009). It therefore seems worthwhile to narrow down PHT stepwise to monitor the residual dose to assess the results for the sample in question (**Supplement 2, Fig. S2.11; Supplement 3, Fig. S3.17**). Therefore, PHT was reduced to 200 °C for the dune sample HDS-1849 and the fluvial sample HDS-1827.

As charge-transfer from REGs may also influence the results, for HDS-1849, we performed zero-dose tests with different dose-response curves (100 s/10.6 Gy *versus* 50 s/5.3 Gy dose-response curve) (**Supplement 3, Fig. S3.1–S3.3, S3.15**), which, however, gave equally poor results, as the PHT to 320 °C was too strong. For feldspar from the Nebraska Sand Hills Buckland et al. (2019) reported that resetting of the latent luminescence signal to a quasi-zero background occurred only after ca ten days of daylight bleaching. Therefore, we increased the bleaching time to 12 h with the solar simulator, but could
not reduce the residual dose with the PET-IRSL SAR protocol at PHT 320 °C (**Supplement 3, Fig. S3.16**). We assume that the determined residual dose results from thermal transfer of unbleachable TL traps into the (PET) IRSL trap(s). Such thermal transfer could also explain that $IR_{1st}$ and low-temperature PET IRSL may underestimate a given dose (cf. **Fig. 6a–6b**), if the charge transfer leads to the occupation of local/near-neighbour centers during preheat, which during the subsequent (PET) IRSL readout are no longer present.


*Hotbleaches*

As PET-IRSL at $PET_{max}$ is not prolonged but – like a SAR cutheat – shut off once the target temperature is reached, PET-IRSL SAR requires hotbleaches (here BLSL or GRSL for 250 s at $PET_{max}$) after both PET-IRSL readouts of a SAR cycle.
Without hotbleaches the De curves show a notable downward bend at the end of the high-temperature PET-IRSL range (**Supplement 3, Fig. S3.6, S3.7**). IR seems to work equally well for HBL stimulation as GRSL or BLSL (**Supplement 3, Fig. S3.4a, S3.5a**), but was not further investigated.

$pIR_{1st}IR_{2nd}$ and MET protocols often include strong hotbleaches at the end of each SAR cycle, e.g., 280 °C for 100 s
(Fu and Li, 2013) or 50 s (Buckland et al., 2019), despite comparably low PHT of 200 °C. But PET-IRSL does not perform





well if HBL above the PHT are applied, often resulting in a given dose that is overestimated along the complete low- and high-temperature PET-IRSL range (cf. **Supplement 3, Fig. S3.8**). Therefore, HBLs need to be applied carefully **(1)** in each SAR half-cycle and **(2)** with restricted temperature (here equal to $PET_{max}$).

740          *The maximum readout temperature – $PET_{max}$*

IRSL readout at up to 30 K below the preheat temperature has been found acceptable in earlier studies applying $pIR_{1st}IR_{2nd}$ or MET SAR protocols to feldspar (Buckland et al., 2019; Fu and Li, 2013; Roberts, 2012). In this respect PET-IRSL SAR performs similar (**Fig. 7b**, **Fig. 8b**). It seems worthwhile to maximize $PET_{max}$ to sample a possibly stable

luminescence signal at higher temperature (**Fig. 8**). We assessed $PET_{max}$ 170 °C after PHT 200 °C as appropriate for HDS-1849 while a residual dose of ca 3 s/0.3 Gy for 180 °C was regarded as too large for natural doses ≤15 s/≤1. 6 Gy (see **Fig. 18**).

### 7.1.3     a-value and g-value determination

750          *a-values*

PET-IRSL SAR measurements on powdered extracts of HDS-1776 and HDS-1849 reported a-values consistent with published literature, underscoring the nature of the alpha-efficiency as a physical property of the dosimeter. However, for strong thermal (pre-)treatment (here PHT 320 °C, $PET_{max}$ and HBL 280 °C), it should be verified if the material carries a

residual dose which might erroneously increase the a-value. Two possible approaches are either subjecting additional aliquots to a zero-dose test to determine their background, which may be subtracted if necessary, or applying the alpha dose to aliquots, which have already undergone PET-IRSL SAR cycles, annealing any residual dose sufficiently.

          *g-values*


It may be relevant to consider anomalous signal loss typical of feldspar (Wintle, 1973) to avoid age underestimation, and to correct for anomalous fading (Huntley and Lamothe, 2001; Kars et al., 2008; Lamothe et al., 2003). An overview and evaluation of the various fading correction models is given by King et al. (2018) and Riedesel (2025). We applied the model by Huntley and Lamothe (2001) to test, in a few selected cases (see **section 7.3.2–7.3.3**), the plausibility of the De-value results

with the PET-IRSL approach for the fluvial sample HDS-1827 and the dune-blowout sample HDS-1849. Elaborate fading corrections are not part of the present study.





g-value curves, as illustrated in **Fig. 10** and **Supplement 5 (Fig. S5.3.1, S5.4.1, S5.5.1),** help to assess potential age underestimation. We did not determine g-values for the samples we studied with conventional $pIR_{1st}IR_{2nd}$ and MET protocols

for direct comparison with the PET-IRSL g-values. However, the g-values are in the range known from other IR-stimulated feldspar measurements. For HDS-1849, the g-values of the early portion of the low-temperature PET-IRSL range and at the end of the high-temperature range compare within error margins to those published by Buckland et al. (2019) for another sample from the Nebraska Sand Hills ($3.7 \pm 0.5$ % per decade for $IR_{50}$; $1.0 \pm 0.5$ % per decade for $IR_{170}$). The data curves of all three samples illustrate the trend of decreasing g-values with increasing readout temperature, as would be expected from

the general model of feldspar luminescence (Jain and Ankjærgaard, 2011), as an increasing number of localized electron holes are used up with progressing measurement time, allowing more stable electron-trap to electron-hole recombination. Furthermore, the g-values of the loess sample HDS-1776, which had undergone PHT 320 °C, show – especially in the very early low and the late high-temperature PET-IRSL range – lower g-values than the fluvial sample HDS-1827 and the dune sample HDS-1849, which had been subjected to lower PHT at 200 °C.


Nevertheless, the strongly upward trending g-values of the first ca four De integrals of samples HDS-1827 and HDS-1848 are unexpected (cf. **Fig. 10 b–c**, **Supplement 5, Fig. Fig. S5.4.1, S5.5.1**). Although it is possible that the earliest stimulated electrons might be responsible for recombination with particularly near/localized centers and therefore a highly unstable IR signal, especially after mild PHT at 200 °C, we cannot exclude an unrecognized measurement artifact. Such effects

as a cause for unusually high g-values in the context of $pIR_{1st}IR_{2nd}$ measurements have been hypothesized in the literature (Thiel et al., 2011). Kadereit et al. (2020, GChron discussion, Final response) reported unusually high g-values that did not follow the usual logarithmic downward trend at up to ca 0.5-1 h after laboratory irradiation. Buckland et al. (2019) measured an unexpected g-value of $2.02 \pm 1.09$ % per decade on quartz from the Nebraska Sand Hills, from where sample HDS-1849 originates. A closer inspection of the boundary parameters of fading measurements, not only those of the PET-IRSL SAR

approach, seems to be desirable.

In **Supplement 5 (Fig. S5.3.2, S5.4.2, S5.5.2)** g-value curves were plotted next to De-value curves of the same aliquot on which first De values and thereafter the g-values had been determined. It is imaginable that the number of aliquots that show a De plateau after fading correction may be increased by individual aliquot-wise fading correction. This could be

possible, e.g., for the five aliquots of the loess sample HDS-1776 from the LPS Baix, which show modest raising and lowering of the De-value and g-value curves in opposing directions. The same could be possible for the Nebraska dune sample HDS-1849. However, the very early ca five g-value data points protruding at the beginning of the low-temperature PET-IRSL g-value curve would also let the beginning of the flat PET-IRSL De-value curves poke out after fading correction (cf. **Supplement 5**, all aliquots of HDS-1849 in **Fig. S5.5.2** and aliquot 3 of HDS-1827 in **Fig. S5.4.2 (c-1, c-2)**). This effect

questions the accuracy of the very early g-value data points.



For the fluvial sample HDS-1827, De-value differences are too large and g-value differences too small to explain the shape of the De-value curves mainly by different signal loss. For this sample, insufficient bleaching efficiency at the bottom of the riverbed and partial bleaching of the feldspar grains seem the most likely scenario.


Mean g-value curves derived from a set of several aliquots (five in this study; **Fig. 10**) of a sample may look more simplified than the more variable data curves of each aliquot (**Supplement 5, Fig. S5.3.1, S5.4.1, S5.5.1**). Presently, the number of g-values generated with PET-IRSL SAR is limited and does not allow for deciding whether mean g-value curves based on a set of several aliquots should be used for age correction or whether signal fading is appreciably diverse to require

unique fading rates to be calculated for specific aliquots. This issue, however, concerns all IR-stimulated dating techniques. Large data sets would be necessary to address this issue appropriately. However, fading measurements are time-consuming. Therefore, for dating studies, it might be acceptable to extract the most stable signal, as derived from the g-value curve, and accept some potential age underestimation. Mean g-value curves, as in **Fig. 10**, may provide an idea of the potential magnitude of signal loss.


While all measurements showed larger g-values for the early low-temperature range than for the late high-temperature range, each sample provided aliquots which, for the De measurements, showed end-to-end PET-IRSL De-value plateaus which appear to contradict the finding of differing g-values. However, Riedesel et al. (2021) found that in contrast to multi-phase feldspar, single-phase feldspars show little fading of the blue IR-stimulated emission. Therefore, it appears possible that some

aliquots may contain low-fading feldspar dominating the blue emission. The search for aliquots showing a continuously flat PET-IRSL De-value curve may therefore be a worthwhile strategy to extract aliquots suited best for dating, especially as these aliquots also lack indications of partial bleaching. This strategy corresponds to the MET-plateau approach used for the luminescence dating of glacial sediments by Bateman et al. (2025).

**7.1.4    De-value curves reflected against PBL tests and De measurements**

The PET-IRSL De-value curves provide information on the pre-burial bleaching histories of a sample and, likely, the fading history of a sample since deposition. Our experiments of partial bleaching (PBL tests) on sample HDS-1776, both with white and monochromatic light, and on the two other samples with very short white-light bleaching, show the influence of

partial bleaching on the shape of the PET-IRSL De-value curve. However, the effect of signal fading on the shape of a sample´s PET-IRSL De-value curve was not investigated. Therefore, at present, the PET-IRSL De-value curves observed on aliquots carrying the natural luminescence signal (see **Fig. 16–18**) can be interpreted only with respect to the results of the PBL tests. The true picture is likely more complicated and should, in future studies, consider also changes due to signal fading.





Using the De-value curves of the DRTs and PBL tests as a first reference data set, different degrees of bleaching can be recognized (see **Figs. 4–5, 6c–d, 12–13, 16–18**). We propose a differentiation between **(1)** De-value curves showing a continuously flat plateau and **(2)** De-value curves that deviate from such a plateau. Among the latter, differing shapes are observed: **(2a)** Data curves that show variable signals followed by a distinct plateau at high-temperature PET-IRSL. In this case a long plateau indicates very inefficient bleaching, and the De-value plateau may reflect the palaeodose corresponding to

a previous (penultimate or earlier) sediment reworking event. **(2b)** An initial downward bend of the De-value curve (initial "hook") is also suggestive of very insufficient bleaching. Such a hook, especially when combined with a long plateau in the high-temperature PET-IRSL range, indicates insufficient bleaching during the last reworking. **(2c)** A shortened plateau in the high-temperature PET-IRSL range, cut backward at the plateau edge in the direction of increasing readout temperature, points to somewhat more complete but insufficient bleaching. The De values at the far end of the high-temperature PET-IRSL range

represent a minimum of the palaeodose associated with an earlier event of sediment reworking. These interpretations related to scenarios **2b** and **2c** are only possible if PHT, $PET_{max}$ and HBL had been adopted carefully. Appreciably high thermal SAR parameter values will likely lead to De values from the low temperature PET-IRSL which underestimate the true palaeodose and a plateau-like bulge of De values around the PET-peak which may meet the expected paleodose (cf. **Fig. 6a, 6b; Supplement 3, Fig. S3.1; Supplement 2, Fig. S2.1**) or may overestimate the palaeodose likely by the size of the residual dose

recovered with the same parameter values with a zero-dose test (**Fig. 7a**). To recognize a shortened plateau, the length of an intact plateau of a sample must be known (e.g., from a PBL test after red-light bleaching), as these may have variable lengths (e.g., longer for HDS-1776 and HDS-1827, shorter for HDS-1849). **(2d)** A sequence of steeply increasing De values of the high-temperature PET-IRSL up to $PET_{max}$, from which the edge of the plateau has completely disappeared (**Fig. 4b, 12b, 13e**), points to progressed sample bleaching during the last event of sediment reworking, with De values giving a maximum dose

for the last event. **(2e)** A flatter slope of De values of the high-temperature PET-IRSL indicates a further increase of bleaching (**Fig. 12c, 13f**). The De values may be helpful in constraining the timing of the last episode of sediment reworking. **(2f)** Conforming to **(1)**, this (semi-)plateau of De values over the complete IR-stimulated range of the custom curve reflects the time of the last reworking when bleaching reduced both the early, likely more easily bleachable IR-signal, and the later recorded and likely less bleachable IR-signal sufficiently (**Fig. 4a, 12d**). A long end-to-end plateau of De values indicates

thorough resetting of the latent IR-signal, increasing the reliability of the De and age determination.

## 7.2 Discussion of first tentative De estimates

The De-value curves of the natural palaeodoses of the sediment samples show comparable patterns as those obtained from laboratory induced doses. Moreover, the De-value curves appear to reflect different environments of bleaching intensity

suggesting: best bleaching conditions for the dune sample HDS-1849 with dominantly flat and closely spaced De-value curves (**Fig. 18**); the poorest bleaching conditions for the fluvial sample HDS-1827 with wider spaced and variably inclined De-value



curves (**Fig. 17**); and some indication of post-sedimentary reworking for the loess sample HDS-1776 (**Fig. 16**), as suspected by Pfaffner et al. (2024; 2025, in press).

**7.2.1    De values and ages HDS-1776**

Most De-value curves of HDS-1776 show somewhat lower De values at the beginning of the low-temperature PET-IRSL range (ca 1250–1550 s/150–186 Gy) as compared to the later portion of the high-temperature PET-IRSL range (1450–1750 s/174–210 Gy; center ca 1650 s or 198 Gy, respectively). This value is ca 12 % below the central dose of 220.12 ± 12.54 Gy determined for $pIR_{60}IR_{225}$ on feldspar coarse grains by Pfaffner et al. (2025, in press). However, the De determination
of Pfaffner et al. (2025, in press) is not based on a calibration with FI calibration quartz. Differences of 10 % or more between different calibration quartzes have been reported (e.g., Kadereit and Kreutzer, 2013; Richter et al., 2020). The De-value plateau of the high-temperature PET-IRSL range of MS_47b (**Fig. 16a**; mean of De interval 90–115: 1841.6 s/221 Gy, respectively) compares to the central dose of Pfaffner et al. (2025, in press), providing an age of ca 55 ka when using the dose rate of 4.01 ± 0.21 Gy/ka given by Pfaffner et al. (2025, in press). This suggests that the sediment in which the 5 Bw2 palaeosol is
developed was deposited in Greenland stadial (GS)15, or earlier.

**Fig. 16b** shows the results of four aliquots that exhibit a PET-IRSL De-value plateau, two representing an upper level around ca 1670 s (201 Gy) possibly corresponding to ca 50 ka or Heinrich (H) 5 event, respectively. Sediment reworking during that cold spell appears likely. Two aliquots of the SA measurements exhibit a lower plateau of De values around
ca 1515 s (ca 182 Gy; ca 45 ka). The results of the SG measurement MS_45c also conform to that lower level (**Fig. 16c**). Bleaching of the mineral grains during GS12 appears possible. Pfaffner et al. (2025, in press) attributed the formation of the prominent 5 Bw2 palaeosol remains to a succession of warm periods, from Greenland Interstadial (GIS) 14 to GIS12/11. The results gained with the PET-IRSL SAR approach corroborate that interpretation but appear to provide insight into the evolution of the remains of the palaeosol horizon on the part of the chronometry. Similar to Pfaffner et al. (2025, in press), no fading
correction was applied. However, plateau-like De-value curves seem to suggest that the mineral grains on the aliquot had neither been partially bleached prior to deposition nor do they exhibit relevant anomalous fading of the PET-IRSL signal (cf. Bateman et al., 2025). This assumption needs to be substantiated by future studies.





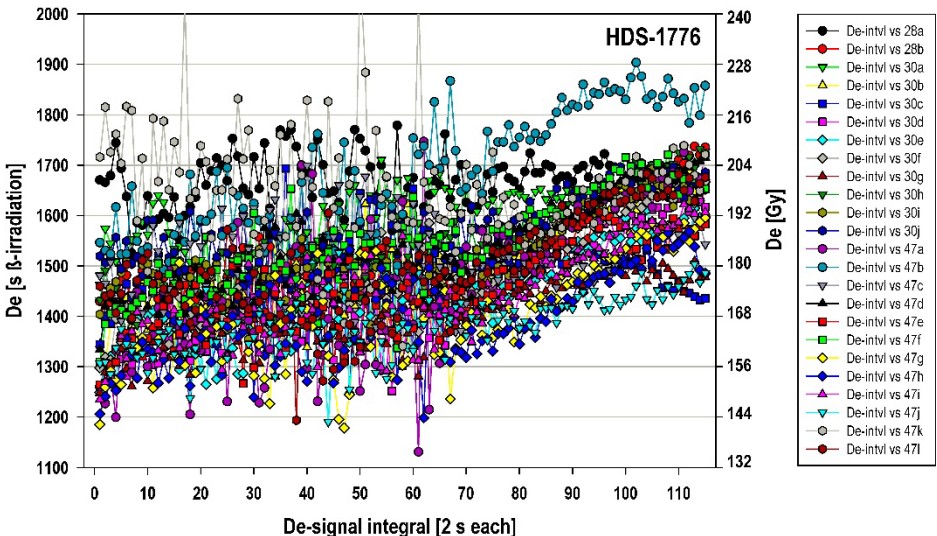

**(a)**

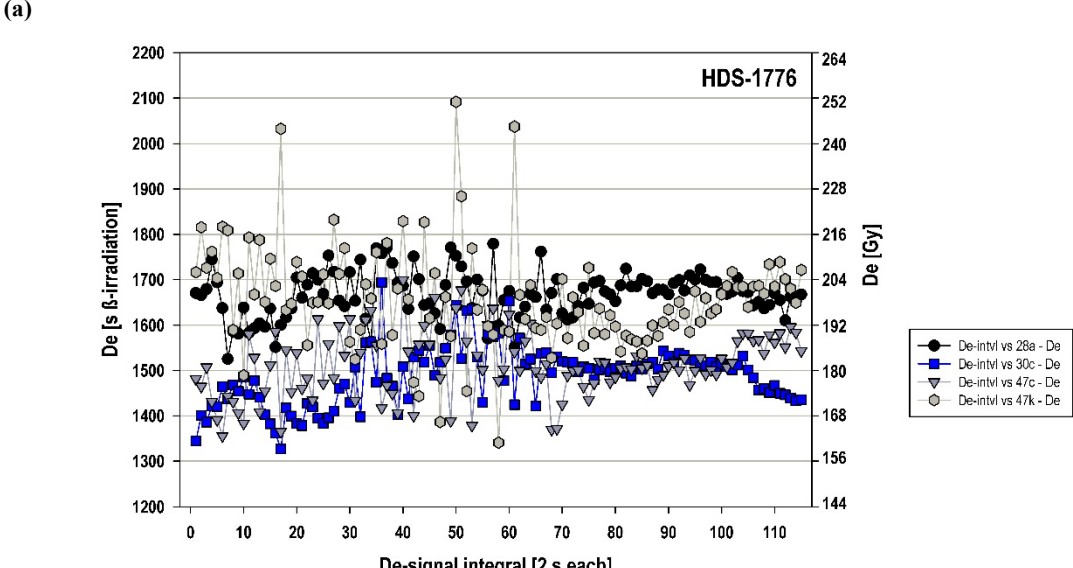

**(b)**



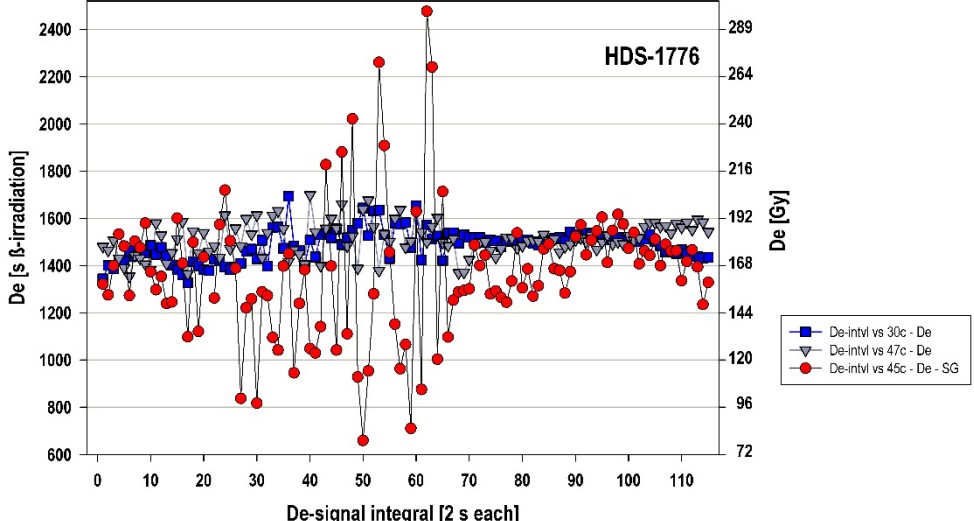

**(c)**

**Figure 16.** Compilation of PET-IRSL De measurements for the sample HDS-1776 from the LPS Baix. **(a)** All 24 multi-grain (MG) aliquots (MS_28a–28b; MS_30a–30j; MS_47a–47l). **(b)** MG aliquots exhibiting a plateau of De values. **(c)** MG aliquots suggesting a plateau of De values around ca 1500 s (180 Gy; lower plateau) and a De-value data curve of a SG aliquot (MS_45c), allowing De determination.

**7.2.2    De values and ages HDS-1827**

HDS-1827 shows the strongest scattering of De values from the beginning of the low-temperature PET-IRSL range, where values range between ca 180 s/21 Gy and ca 300 s/35 Gy (factor of ca 2), to the end of the high-temperature PET-IRSL range, where most values range between ca 240 s/28 Gy and ca 450 s/53 Gy (again a factor of ca 2). Two aliquots reach values

of ca 850 s/100 Gy, reinforcing the apparently insufficient bleaching conditions. In addition, several aliquots exhibit an initial downward bend, some with a plateau-like portion in the late high-temperature PET-IRSL range, which both may point to very modest partial bleaching, perhaps in turbid water. As the sample originates from the bottom of a palaeo-riverbed, this result is not surprising, but the information is conveyed by the shape of the De-value curves in **Fig. 17a**.

**Fig. 17b** shows the three aliquots providing the lowest dose of the late high-temperature PET-IRSL (ca 270 s/32 Gy). While two aliquots show comparably steep De-value gradients, one aliquot shows a data curve that may be interpreted as a plateau, especially if considering the differing g-values along the 230 s readout time (**Fig. 10b**). As high groundwater levels prevailed at the site until the end of the 18[th] century, water-filled pores for most of the time since sediment deposition is the



most likely scenario. Assuming a ratio of 1.5 for wet-sample weight to dry-sample weight, the annual dose rate is, depending

on the potassium content of the feldspar, assessed as 2.2–1.8 Gy/ka (cf. **Supplement 6, Fig. 6.6**). 270 s/31.8 Gy might –
depending on the feldspars´ potassium content – point to an event of fluvial reworking with sufficient signal resetting in
Lateglacial times, ca 14.5–17.7 ka. Correcting, as a trial, for fading using a g-value of 1.4 % per decade (cf. **Fig. 10b**, integral
71-110) would provide, a slightly older age around ca 16.4–20.1 ka. The initial ca 250 s/29.5 Gy result in ca 13.4–16.4 ka
depending on the assumed potassium content. Fading corrected with a g-value of 2 % per decade (cf. **Fig. 10b**, integral 6-20),

these would become ca 16.0–19.7 ka, very similar to the corrected ages for the high-temperature PET-IRSL calculated with a
loss of 1.4 % per decade. The results are coherent.

The lowermost, for fading uncorrected De values from the tip of the De-value curves (**Fig. 17c**) would give a dose of
ca 21.2 Gy, resulting in an uncorrected age of 9.6–11.8 ka. The sediment possesses a framework of pebbles (Engel et al., 2022)

and finer material, including the fraction 125–212 μm used for PET-IRSL SAR measurements, may have been washed into
the voids between the gravelly components from above during later events. [14]C-dating suggests that Early-Holocene material
(ca 7.8 ka cal BP) from the time when the Bergstraßenneckar was likely last active at the Schäffertwiesen site directly overlies
the Lateglacial deposits (ca 11.2 ka cal. BP) (Engel et al., 2022).

Applying tentative fading correction with a g-value of 3 % per decade (cf. **Fig. 10b**, first two integrals) to the
preliminary ages derived from the lowermost De values would, again depending on the assumed potassium content, result in
Lateglacial ages of ca 12.6–15.5 ka. This scenario would also be plausible when considering the 11.2 cal BP age from the
stratigraphic layer above as independent age control (Engel et al., 2022, Fig. 3b).

Obviously, a complicated history of fluvial deposition cannot be derived from one sample with a limited number of
aliquots. Our attempts at interpretation mainly serve to illustrate the potential that quasi-continuous De-value curves may
provide for future studies. Basically, the results are plausible. Using them for the originally intended purpose of a screening,
one might conclude that the bleaching conditions for sample HDS-1827 are not optimal for luminescence dating, and, if better
samples are not available, SG pIR$_{1st}$IR$_{2nd}$ should be applied.




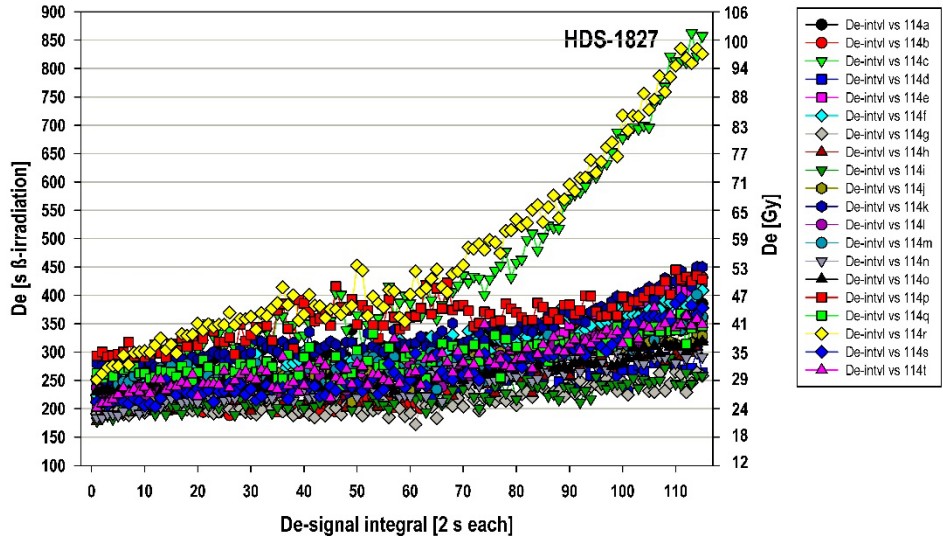

**(a)**

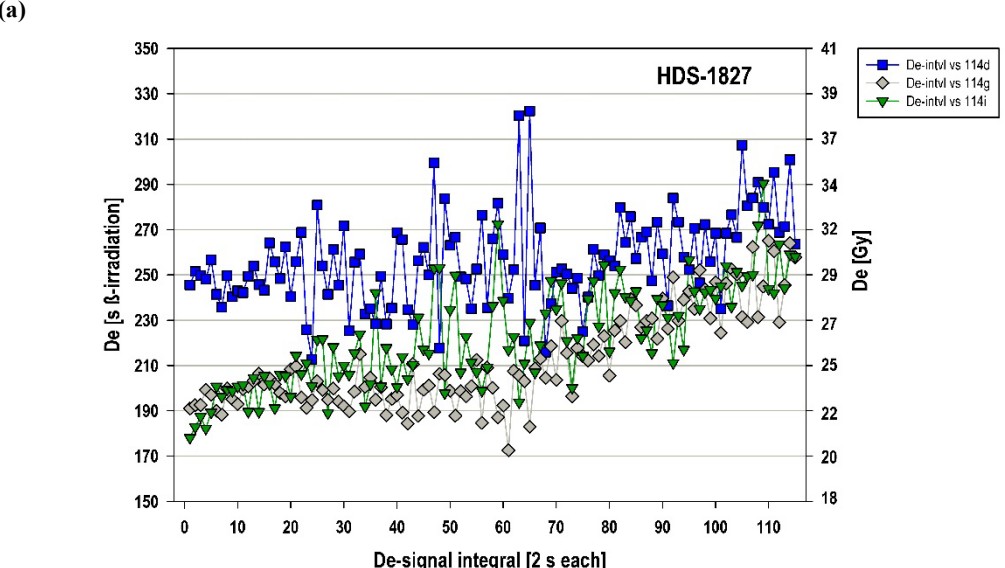

**(b)**





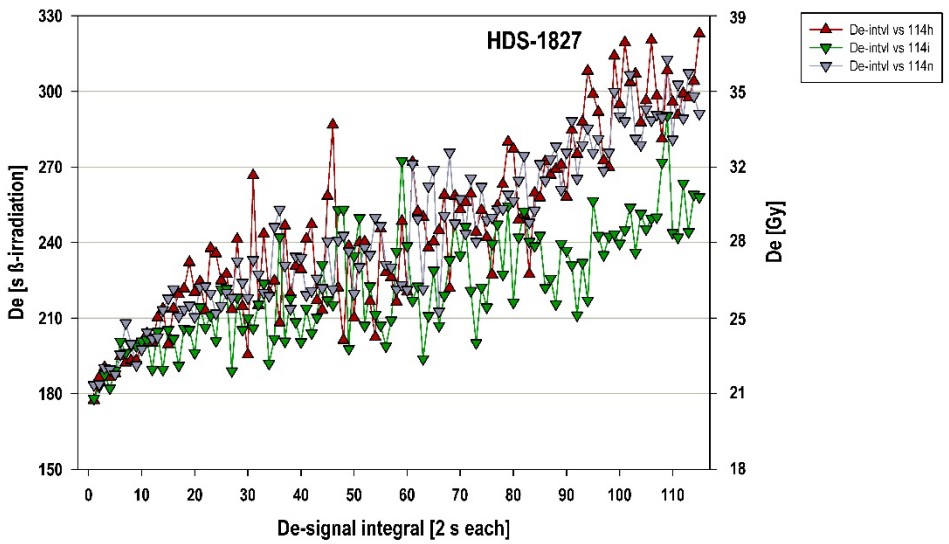

**(c)**

**Figure 17.** Compilation of PET-IRSL De measurements for the sample HDS-1827 from the bottom of the riverbed of the palaeoriver Bergstraßenneckar in southwestern Germany. **(a)** All 20 aliquots (MS_114a–114t). **(b)** Aliquots delivering the three smallest Des in the late high-temperature PET-IRSL range. **(c)** Aliquots providing the smallest Des for the tip of the low-temperature PET-IRSL range.

### 7.2.3    De values and ages HDS-1849

Sample HDS-1849 from the Nebraska Sand Hills appears to be well bleached. The De-value curves in **Fig. 18a** mostly
show plateau-like shapes clustering closely between ca 12–14 s (1.3–1.5 Gy) in the early low-temperature PET-IRSL range and ca 12–15 s (1.3–1.6 Gy) in the late high-temperature PET-IRSL range (ignoring the late low-temperature PET-IRSL range in which outliers due to the low signal strength complicate the overall picture). Extracting the highest (**Fig. 18b**) and the lowest (**Fig. 18c**) measured De values (mean over all 115 De-signal integrals) helps clarify the picture further. The De-value curves in **Fig. 18b** start at the lower end in the range 13–14 s (1.4–1.5 Gy) and end around 14–15 s (1.4–1.6 Gy). Thus, despite only
a minor increase of De of ca 7 %, the aliquots with the highest palaeodose are suggestive of some degree of partial bleaching. None of the De-value curves exhibits an initial downbending. If partial bleaching is the reason for the deviation from a simple plateau, signal resetting was probably adequate prior to deposition. However, signal fading may also explain the differences between the early and late De values. Assuming a representative ratio of wet to dry weights of 1.01 for this sample, would, depending on potassium content, produce ages of ca 0.43–0.49 ka (lower end of De-value curve) to 0.46–0.52 ka (upper end).



Applying tentative fading corrections with g-values of 4 % per decade (lower end; integrals 1–5) and 1.8 % per decade (upper end; integrals 90–115) (cf. **Fig. 13c**) results in similar ages of ca 0.58–0.67 ka (lower end) to ca 0.53–0.60 ka (upper end).

In contrast to **Fig. 18b**, the aliquots delivering the lowest De values do not show such an increasing trend (**Fig. 18c**). De values begin in the range 12–13 s (1.3–1.4 Gy) in the low-temperature PET-IRSL range and – apart from some scatter –
end in that range in the high-temperature PET-IRSL range. The corresponding ages are 0.40–0.45 ka. As the aliquots show De plateaus, they might not show signal fading (cf. Bateman et al., 2025). Alternatively, they could show similar signal loss in both the low and the high-temperature PET-IRSL range, an assumption not suggested by the fading measurements (cf. **Supplement 5, Fig. S5.5.1**). Also, the fading measurements on the aliquots showing plateau-like De-value curves (MS_32g, MS_32j; cf. **Supplement 5, Fig. S5.5.2(b-1)/(b-2) and (d-1)/(d-2)**) exhibit higher g-values at the start than at the end of the
data curves. The lower De values (ages) of the aliquots compiled in **Fig. 18c** could possibly also be explained by relatively high potassium contents and/or post-depositional bleaching due to bioturbation.

Anyway, either age, generally in the range ca 0.4–0.6 ka, point to a more recent date of sediment reworking than the quartz age of ca 1 ka (2.26 Gy) given by Kreutzer (unpublished data) although quartz is regarded to bleach faster than feldspar
(Godfrey-Smith et al., 1988). For feldspar from the Nebraska Sand Hills Buckland et al. (2019) found that the residual dose bleaches to zero only after ca 10 days of exposure to sunlight, but they reliably dated deposits of known age of only a few years. Individual quartz aliquots of HDS-1849 Kreutzer (unpublished data) gave De values as low as 1.1–1.7 Gy corresponding roughly to 0.55–0.85 ka and overlapping with the tentatively calculated PET-IRSL feldspar ages. Yet, for the quartz ages one would assume that De values on the low end of the distribution are adversely impacted by bioturbation. We cannot explain the
difference between the palaeodoses stored by quartz and feldspar. However, we measured a few aliquots of feldspar with a conventional pIR$_{50}$IR$_{170}$ protocol on the same reader (LR04, "Colour") as the PET-IRSL SAR measurements (cf. **Supplement 6, Fig. S6.3**). The protocol was tested with DRTs with differing NRMs and a zero-dose test, which all performed well. As a result, the De measurements on the aliquots with the natural signal corroborated the results of the PET-IRSL SAR protocol (ca 12.5–14.2 s; 1.33–1.51 Gy). If quartz gives the true age and feldspar underestimates the age of the blowout-
deposit, this must be a not yet understood issue of the feldspar and may not be attributed to a malfunctioning of the PET-IRSL SAR protocol. In this context, findings of Mey et al. (2023) appear interesting who observed better bleaching of feldspar in a turbid subaqueous suspension of feldspar and quartz in which the short-wave (UV, blue) component of the daylight was stronger attenuated than the long-wave (red, near IR) fraction. It is possible that aeolian re-working in a subaerial dust and sand cloud may have a similar effect (cf. Anton et al., 2012), and bleaching experiments in the future may be helpful for
interpreting these unexpected results.



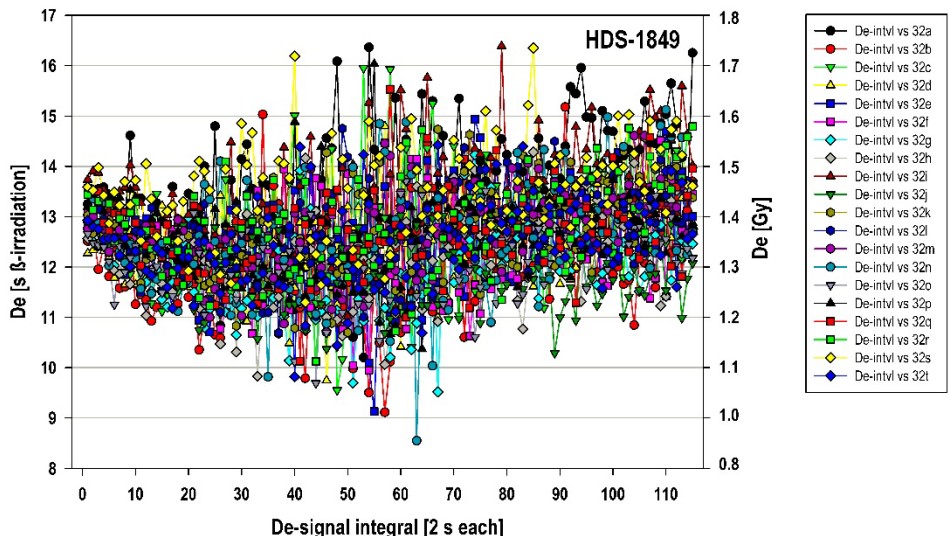

**(a)**

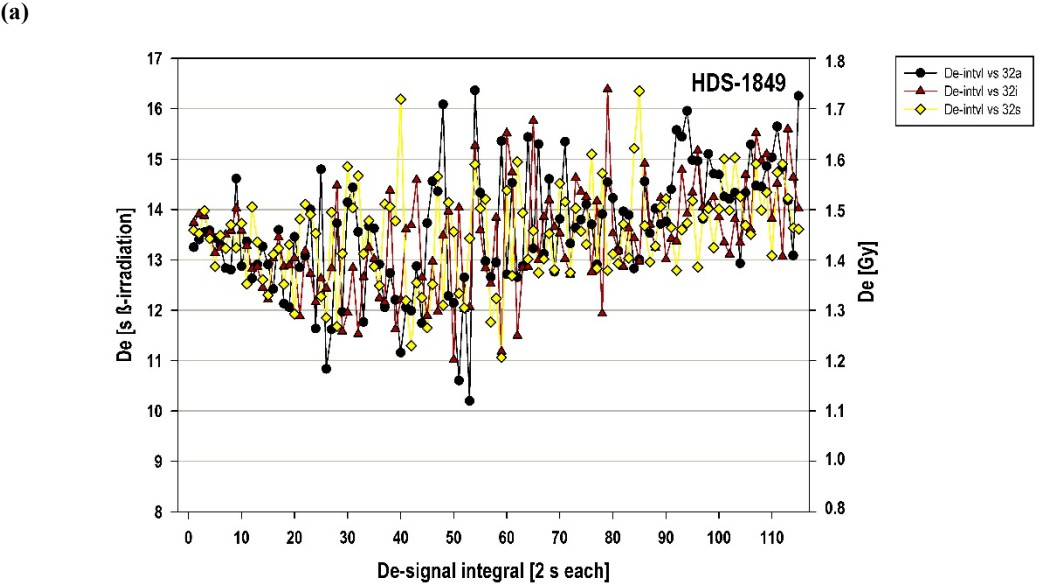

**(b)**





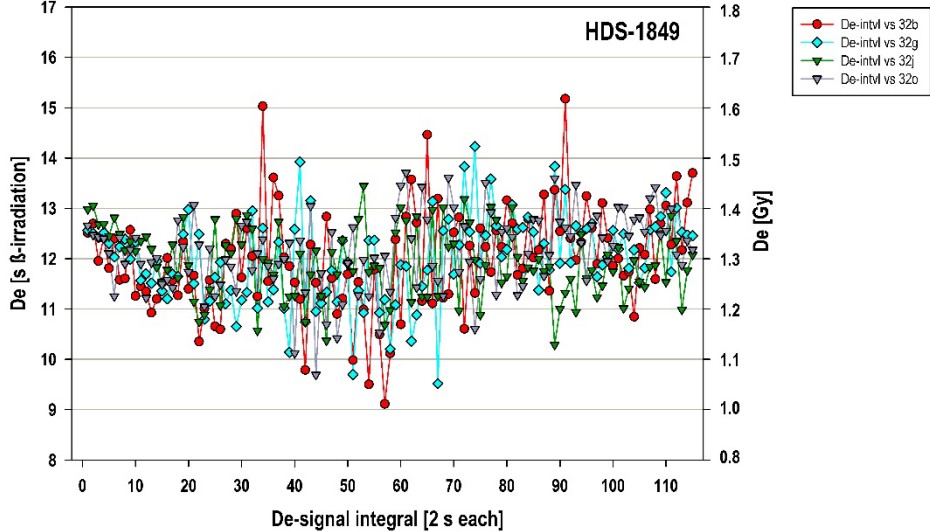

**(c)**

**Figure 18.** Compilation of PET-IRSL De measurements for the sample HDS-1849 from the dune sample in the Nebraska Sand Hills. **(a)** All 20 aliquots (MS_32a–32t). **(b)** Aliquots delivering the two highest Des (mean of all 115 De intervals). **(c)** Aliquots providing the four lowest Des (mean of all 115 De intervals).

# 8   Conclusion

We introduced the progressively elevated temperature (PET) IRSL SAR approach that represents a further
development of IRSL, post-IR IRSL and MET techniques for feldspar. The method extracts a comparably stable luminescence signal by sampling sequentially, first, less stable and thereafter the remaining more stable IRSL, that is isolated for paleodose estimation. Whereas the established techniques determine De values for two ($pIR_{1st}IR_{2nd}$) or several (MET) IR-stimulation temperatures, PET-IRSL provides a quasi-continuous De-value curve over the investigated temperature range (here up to 280 ºC), recorded over 230 s, resulting in a sequence of 115 De data points. We assume that PET-IRSL samples the same IR-
trap(s) as the established techniques, but apparently with reduced electron recapture and recycling during the progress of a SAR measurement, as earlier observed for $pIR_{1st}IR_{2nd}$, which is likely the result of lacking cool-down and IR-switch-off steps that are intercalated in the traditional SAR protocols. The quasi-continuous PET-IRSL De-value curves provide detail on partial bleaching and anomalous signal fading. In addition, PET-IRSL allows for the derivation of a-value and g-value curves. We presented the first plausible De values obtained with PET-IRSL and tentative a-value and g-value estimates which compare
favorably with those calculated from $pIR_{1st}IR_{2nd}$ and MET, indicating that the PET-IRSL approach performs reliably.



Differing shapes of De-value curves may include consistent plateaus over the complete low- and high-temperature PET-IRSL signal range, or an initial downward bending, or a steady high-temperature PET-IRSL De-value increase with progressing measurement time with either a relatively modest or steeper gradient, or a plateau in the (late) high-temperature PET-IRSL range. These resultant curves may be useful in evaluating the specific bleaching history of a sample as suggested by partial bleaching (PBL) tests in the present study.

The investigated sediment samples, which all show comparable characteristics under PET-IRSL, are of different ages (Pleniglacial to historic times) and originate from differing palaeoenvironments (far transported aeolian loess, fluvial riverbed deposit, locally reworked dune blowout) and geographical regions (southern France in Mediterranean Europe, Germany in central Europe, Nebraska in central USA). The specific features of a PET-IRSL curve **(1)** starting with a shine-down during the low-temperature PET-IRSL readout (here 120 s at 60 °C), **(2)** followed by a rising curve of luminescence signals in the high-temperature PET-IRSL range, and **(3)** forming a dome-like PET-peak ca 200–220 ºC were reproducible on chemically well-defined feldspar specimens. In summary, PET-IRSL appears to have the potential to be developed into a dating method supplementing the family of hitherto established feldspar luminescence-dating techniques. Even if one does not intend to apply PET-IRSL for feldspar dating, sample analysis by PET-IRSL SAR may be useful in evaluating a sample's bleaching and signal-fading history. Mathematical deconvolution of the data curves, which were evaluated by visual inspection in the present study, might, in future studies, improve the interpretation of a sample's luminescence characteristics and its depositional history.

***Data availability.*** Data will be made available on Zenodo in compliance with the journal policy.

***Author contribution.*** Author contribution. AK: writing-original draft; conceptualisation; formal analysis; methodology. MS-G: writing – review and editing; investigation; methodology. SK: writing – review and editing; methodology; validation; software; MC: writing – review and editing; software; CS: writing – review and editing; validation; PRH: writing – review and editing; validation.

***Competing interests.*** The contact author has declared that none of the authors has any competing interests.

***Acknowledgment.*** We thank Jutta Asmuth for managing all the lab work so smoothly. Volker Schniepp helped with figure 1 and 3.

***Financial support.*** Sebastian Kreutzer was supported through the DFG Heisenberg programme (no. 505822867). Marco Colombo was financed through the DFG project "REPLAY" (no. 528704761).



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
