# Peer review of "A progressively elevated temperature (PET) IRSL SAR procedure – first experiments and results"

_EGUsphere, 2025_

## Referee Comment (RC2)

**Review of Kadereit et al. A progressively elevated temperature (PET) IRSL SAR procedure – first experiments and results**

The manuscript by Kadereit et al. presents a new feldspar luminescence dating protocol, which is based on a SAR IRSL protocol and follows the idea of a post-IR IRSL or MET-IRSL protocol. However, instead of performing individual IRSL measurements at increasing temperatures, the protocol measures the IRSL signal at continuously increasing temperatures, basically by performing a TL measurement while having the IR LEDs turned on. Besides introducing the protocol, the authors also present a range of quality control tests, which are meant to characterise the performance of the protocol.

While I very much appreciate the considerable effort the authors have clearly put into this study, I have major reservations about the current presentation of the results. The authors have performed a very large number of experiments, which is visible by the length of the paper (57 pages), which includes 18 Figures (each of them consisting of multiple sub-figures), and the length of the supplementary material (161 pages). Regardless of the content, I find this paper too long and too complex. I would like to ask the authors to cut the paper considerably and consider splitting it into two papers.

Besides the length, I find the structure of the paper not ideal and I am so far not convinced why this protocol is an improvement to existing measurement procedures. I would like to ask the authors to, when revising their manuscript, consider, which information is really important and how this information can be supported by a selection of clear figures and well-structured text.

Right from the beginning, I was wondering, why no introduction of the physical processes governing the proposed protocol was given and why the authors did not comment on the possibility of phototransfer affecting their signal (see e.g. Wang and Wintle, 2013; Qin et al., 2015; Riedesel and Duller, 2022). This is then addressed in section 6.4. However, the figures are not very clear and zoomed in version would be helpful and a further discussion of the drawbacks of phototransferred TL would be useful. At the end of section 6.4 the reader is left wondering, if one should now be concerned about the possibility of PT-TL or not.

Many figures are presented to illustrate the data, however, most of them look similar and while they of course present the data, the key message each figure is supposed to convey is not directly visible. Regarding the figures I would like to offer the following suggestions:

- Please display all doses in Gy and not in s.
- Currently there is text pasted below each subfigure. I would suggest adding this labelling into each subfigure (text type and sample: for example "Dose recovery test HDS-1849 (PHT 320 °C)")
- I give further comments regarding individual figures below.

***Suggestion of restructuring:***

Overall, I would suggest splitting the paper into two or restructuring it significantly:

(1) One paper or the first part of a paper presenting the concept of the protocol, with an explanation of the physics behind it and the potential influence of PT-TL on the measured signal. It could then also include a sketch-like summary figure which presents the types of De plateaus expected. This would be particularly helpful to be able to follow the

discussion in section 7.1.4. Bateman et al. (2003) present a conceptional, sketch-like figure (their Fig. 1) to illustrate De distributions they would expect for certain bleaching processes. I would envisage something like for the expected De plateaus. It would also be interesting to add a figure showing the variation in peak position.

(2) A second paper or second part of a paper could then evaluate various quality control parameters, such as DRT performance or size of residual dose. Here a few key figures could be shown to support the data. I don't think it is necessary to show 18 figures in the main text. Instead of showing many similar figures, summary figures could be presented.

For these two papers or sections of a paper it is important that they make it clear, why this protocol is an important contribution to the existing suit of measurement protocols. Why should a user deviate from their routine procedures to test this new protocol? What are the advantages?

In many instances throughout the text, the authors use abbreviations or deviate in their description of their experiments from common terminology. This makes it hard to follow the text, as it requires constantly remembering how certain tests and procedures are termed in the current paper. For example: an NRM should be called test dose, a zero-LAB DRT is a residual dose measurement.

***Further comments:***

Throughout the text: Please present all doses in Gy and not in s. I can see that you use both in the figures (two axes) and the text, but I don't see the necessity in this. It adds to the complexity of both, the text and the figures.

Abbreviations: Many abbreviations are used throughout the text. I am particularly confused by the use of LAB, NRM and PBL. NRM is usually referred to as test dose in the literature. I would ask the authors to keep in line with the general use of terms (see also the naming in Murray and Wintle 2000, their Table 1).

Lines 31-40: This is a very long and unnecessary introduction. Is there the possibility to shorten this and target the first paragraph of the introduction to convey the necessity of this study?

Lines 57-58: I disagree in calling band tail hopping "temperature-dependent fading".

Line 62: "Firstly, near-neighbor electron…" There is a preheat prior to the low temperature IRSL step, which also results in electron hole-recombination.

Lines 95-96: I would recommend citing Jain and Ankjærgaard (2011) instead of Riedesel et al. (2019).

Lines 99-128/Section 2.1 (sample material): I would suggest moving the information into a table. Please explain why these samples were chosen, although they don't have any independent age control available.

Line 264: Intensity rather than count data?

Lines 396-399: This is very long and complex sentence.

Line 412: Referring to fading as luminescence decay is rather uncommon.

Line 425: Please change to: All g-values ae in the range of 0.5-4 % per decade, indicating a signal loss over time.

Lines 441-455/Section 6.1: This section could be cut. It comes very suddenly and only conveys the information that the samples exhibit a blue emission. This could have been added to the sample description.

Line 555: What is meant by "late background"? I can see from Figure 3 that late background describes the signal measured after the LEDs have been turned off, but that is the instrumental background and not the signal background.

Line 593: Higher instead of stronger.

Lines 600-614: A figure just showing the peak positions would be useful.

Line 675 and elsewhere: Please do not call them zero dose tests. What is measured is the residual dose remaining in the samples after laboratory bleaching.

Lines 767-774: I cannot follow the reasoning. G-values vary significant between samples.

Line 781: I would say the g-values decrease with De integral?

Line 825/section 7.1.4: I don't understand what the authors mean by partial bleaching test and how they were conducted. I understand this as investigation into the shape of De plateaus during a De determination using the proposed protocol.

Fig. 1: Why is there a secondary y-axis displaying a change in power density? I thought the density was set at e.g. 60 mW/cm2 and not changed during the measurement? I guess the green box is inserted to show the power density? However, with the actual scale on the secondary y-axis it seems like the power density should be changed during the measurement.

Fig. 2: Here especially: Please report all doses in Gy and not in s. You could then also remove most footnotes to make the figure more readable.

Fig. 3: In the legend please use lines instead of points.

Fig. 5: Replace "expected value" by "given dose", you could even add a little label to the red line in the figure. This would also help the reader to immediately spot that this data was gathered by performing a dose recovery test.

Fig. 5 and Fig. 6: These figures are referred to asl NRM tests. I thought they show the results from a dose recovery test and the NRM is the normalisation dose, which is usually called a test dose. Or am I mistaken?

Fig. 7: In the caption you refer to the residual dose tests as zero-LAB DRT. Please refer to them as residual dose measurements/test. There is no dose to recover.

Fig. 9: The orange star is missing in b.

Fig. 16: The overall message these figures should convey is unclear to me. Is it also necessary to show all Figures 16, 17 and 18?

***References:***

- Bateman, MD., Frederick, C.D., Jaiswal, M.K., Singhvi, A.K. Investigations into the potential effects of pedoturbation on luminescence dating. Quaternary Science Reviews 22, 1169-1176, 2003.

- Jain, M., Ankjærgaard, C., 2011. Towards a non-fading signal in feldspar: insight into charge transport and tunnelling from time-resolved optically stimulated luminescence. Radiat. Meas. 46, 292–309.
- Murray, A.S., Wintle, A.G. Luminescence dating of quartz using an improved single-aliquot regenerative-dose protocol Radiation measurements 32, 57-73, 2000.
- Qin, J., Chen, J., and Salisbury, J. B. Photon transferred TL signals from potassium feldspars and their effects on post-IR IRSL measurements. Journal of Luminescence, 160: 1–8, 2015.
- Riedesel, S., Duller, G.A.T. Measuring photo-transferred thermoluminescence from feldspars in post-IR IRSL procedures using a new user defined command for the Risø TL/OSL reader. Ancient TL 40 (1), 2022.
- Wang, X. L. and Wintle, A. G. Investigating the contribution of recuperated TL to post-IR IRSL signals in perthitic feldspar. Radiation Measurements, 49: 82–87, 2013.

Kind regards,

Svenja Riedesel

Cologne, 9th February 2026